# SAT-LDM: PROVABLY GENERALIZABLE IMAGE WA-TERMARKING FOR LATENT DIFFUSION MODELS WITH SELF-AUGMENTED TRAINING

## ABSTRACT

The proliferation of AI-generated images necessitates effective watermarking to protect intellectual property and identify fake content. While existing training-based watermarking methods show promise, they often struggle with generalization across diverse image styles and tend to produce noticeable artifacts. To this end, we introduce a provably generalizable image watermarking method for Latent Diffusion Models with Self-Augmented Training (SAT-LDM), which aligns the training and testing phases by a free generation distribution to bolster the watermarking module's generalization capabilities. We theoretically consolidate our method by proving that the free generation distribution contributes to its tight generalization bound without the need to collect new data. Extensive experimental results demonstrate that SAT-LDM achieves robust watermarking while significantly improving the quality of watermarked images across diverse styles. Furthermore, we conduct experimental analyses to demonstrate the strong generalization abilities of SAT-LDM. We hope our method offers a practical and convenient solution for securing high-fidelity AI-generated content.

## 1 INTRODUCTION

Recent developments in diffusion models, notably commercial models like Stable Diffusion (SD) (Rombach et al., 2022), Glide (Nichol et al., 2022), and Muse AI (Rombach et al., 2022), have revolutionized image generation. These models exhibit exceptional capabilities in generating high-quality and diverse images from textual descriptions, making them valuable tools across a range of domains, such as fashion design (Baldrati et al., 2023) and education (Lee & Song, 2023). However, the ease of generating such images also raises concerns about intellectual property rights and the propagation of fake content, making it imperative to watermark generated content.

As a tailored approach, watermarking technology (Cox et al., 2007) can embed hidden messages into images, facilitating copyright verification and source identification. Traditional post-hoc watermarking techniques (Xia et al., 1998; Zhu et al., 2018) introduce watermarks after image creation, adding extra workflow steps and potentially degrading image quality (Fernandez et al., 2023). In response to these limitations, recent efforts (Fernandez et al., 2023; Xiong et al., 2023; Yang et al., 2024) have shifted towards diffusion-native watermarking, where watermarking is integrated directly within the image generation pipeline. Notable methods such as Stable Signature (Fernandez et al., 2023) and FSW (Xiong et al., 2023) fine-tune the VAE decoder within the latent diffusion model (LDM) as a watermarking module on public image datasets to embed watermarks. While these methods show promise, *they still fall short when applied to real-world scenarios*, often resulting in noticeable artifacts that degrade image quality, as shown in Figure 1. This degradation may stem from the biases in the public datasets, which fail to cover the diverse image styles present in actual use and exceeds the practical ability of generative models, thus leading to a watermarking module with limited generalization ability. A straightforward solution might involve larger datasets encompassing more diverse styles, but this approach demands substantial resources and time, and may also raise concerns about data privacy and copyright Khan & Hanna (2022). Driven by the significance of watermarking effectiveness—the invisibility and robustness of watermarks—a key question arises:

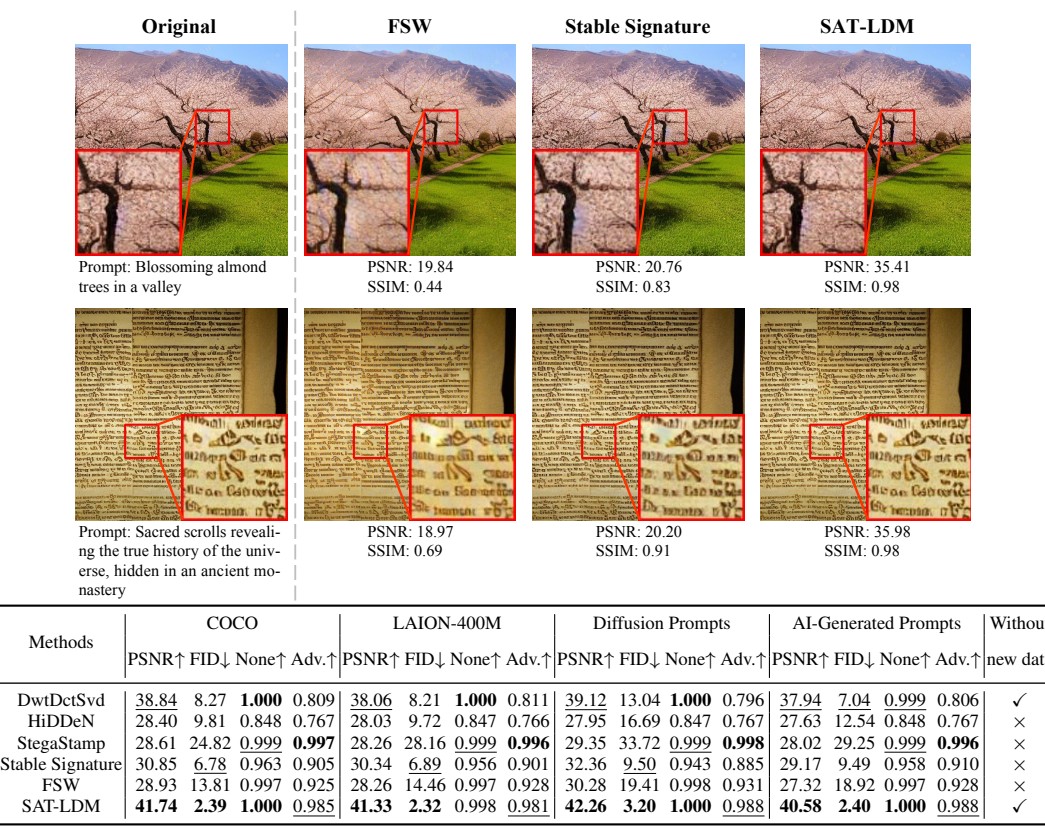

| Methods | COCO | | | | LAION-400M | | | | Diffusion Prompts | | | | AI-Generated Prompts | | | | Without new data |
|---|---|---|---|---|---|---|---|---|---|---|---|---|---|---|---|---|---|
| | PSNR↑ | FID↓ | None↑ | Adv.↑ | PSNR↑ | FID↓ | None↑ | Adv.↑ | PSNR↑ | FID↓ | None↑ | Adv.↑ | PSNR↑ | FID↓ | None↑ | Adv.↑ | |
| DwtDctSvd | 38.84 | 8.27 | **1.000** | 0.809 | 38.06 | 8.21 | **1.000** | 0.811 | 39.12 | 13.04 | **1.000** | 0.796 | 37.94 | 7.04 | 0.999 | 0.806 | ✓ |
| HiDDeN | 28.40 | 9.81 | 0.848 | 0.767 | 28.03 | 9.72 | 0.847 | 0.766 | 27.95 | 16.69 | 0.847 | 0.767 | 27.63 | 12.54 | 0.848 | 0.767 | × |
| StegaStamp | 28.61 | 24.82 | 0.999 | **0.997** | 28.26 | 28.16 | 0.999 | **0.996** | 29.35 | 33.72 | 0.999 | **0.998** | 28.02 | 29.25 | 0.999 | **0.996** | × |
| Stable Signature | 30.85 | 6.78 | 0.963 | 0.905 | 30.34 | 6.89 | 0.956 | 0.901 | 32.36 | 9.50 | 0.943 | 0.885 | 29.17 | 9.49 | 0.958 | 0.910 | × |
| FSW | 28.93 | 13.81 | 0.997 | 0.925 | 28.26 | 14.46 | 0.997 | 0.928 | 30.28 | 19.41 | 0.998 | 0.931 | 27.32 | 18.92 | 0.997 | 0.928 | × |
| SAT-LDM | **41.74** | **2.39** | **1.000** | 0.985 | **41.33** | **2.32** | 0.998 | 0.981 | **42.26** | **3.20** | **1.000** | 0.988 | **40.58** | **2.40** | **1.000** | 0.988 | ✓ |

Figure 1: **Visual comparison of the watermarking methods,** including Stable Signature, FSW, and our proposed SAT-LDM, along with their performance across various datasets. We can observe that Stable Signature introduces a blue-gray spot and FSW exhibits noticeable glare, while SAT-LDM produces more visually appealing results. The size of each test dataset is 1K. "None" and "Adv." represent the average bit accuracy for images without attacks and with adversarial attacks, respectively. The symbol ↑ means higher is better; while ↓ means lower is better. The best-performing method for each metric is highlighted in **bold**, and the second-best is underlined. SAT-LDM effectively handles prompts from public datasets (COCO (Lin et al., 2014) and LAION-400M (Schuhmann et al., 2021)) as well as prompts that better reflect real-world scenarios (Diffusion Prompts (Gustavosta, 2022) and AI-Generated Prompts). See Appendix F for more visualization results.

*Can we train a watermarking module that theoretically performs well across various image styles without collecting new data?*

To this end, we propose Image Watermarking for Latent Diffusion Models with Self-Augmented Training (SAT-LDM), a novel watermarking framework that bridges the gap between effectiveness and generalization across diverse image styles in watermarking. It fundamentally rethinks how watermarking modules in LDM are trained: instead of relying on external datasets that may involve bias, SAT-LDM leverages an internally generated free generation distribution, which aligns closely with the natural conditions under which diffusion models operate. This free generation distribution, formed without specific prompts, ideally mirrors the conditional generation distribution that incorporates all possible prompt-driven outputs. By training the watermarking module on this free generation distribution, we ensure that the model can generalize effectively across a wide range of image styles, without the need for any external data. Furthermore, we theoretically consolidate our method by proving that the free generation distribution contributes to a tight generalization bound, significantly reducing the distributional discrepancy between the training and testing phases. This tight bound guarantees the SAT-LDM can deliver both robust watermarking and high-quality images, offering a practical and convenient solution for protecting AI-generated content.

To evaluate the invisibility and robustness of the watermarks, we generate 10 diverse prompt categories using GPT-4 and create 100 prompts with varying styles for each category, enabling a comprehensive

assessment of the method's generalization across a wide range of image styles (AI-Generated Prompts). Extensive experimental results demonstrate that SAT-LDM not only maintains high robustness similar to previous methods but also significantly improves watermarked image quality across different image styles. Additionally, a detailed experimental analysis provides further empirical support for the validity of our theoretical analysis.

## 2 RELATED WORK

### 2.1 POST-HOC WATERMARKING

Post-hoc watermarking methods involve embedding watermarks into images after their creation and can be classified into three categories: (1) Frequency domain method (Cox et al., 2007) embed watermarks by manipulating the frequency components of image, balancing robustness and complexity. (2) Per-image optimization (Kishore et al., 2021; Fernandez et al., 2022) customizes watermark embedding for each image, allowing for more hidden information but increasing computational demands. (3) Encoder-decoder networks (Zhu et al., 2018; Tancik et al., 2020; Jia et al., 2021) enhance robustness against compression and real-world image transformations, with the option to incorporate targeted adversarial training to further improve watermark robustness against other attacks. However, when applied to images generated by diffusion models, these post-hoc methods introduce additional workflow steps independent of the generation pipeline. This not only increases time overhead but also may degrade image quality (Fernandez et al., 2023).

### 2.2 DIFFUSION-NATIVE WATERMARKING

Diffusion-native watermarking integrates the watermarking process directly into the diffusion model's workflow. This category can be further divided into training-free methods and training-based methods.

**Training-free Methods.** Tree-Ring (Wen et al., 2024) introduces the concept of embedding watermarks in the initial noise during the diffusion process, achieving notable robustness but lacking multi-key identification (Ci et al., 2024b). Subsequent methods enhance this by using improved imprinting techniques (Ci et al., 2024b; Yang et al., 2024). Despite their advancements, these methods can significantly alter the layout of the generated images, which may be undesirable in certain production scenarios (Ci et al., 2024a).

**Training-based Methods.** DiffuseTrace (Lei et al., 2024) hides information in the initial noise by training a dedicated encoder/decoder. WaDiff (Min et al., 2024) and AquaLoRA (Feng et al., 2024) embed watermarks into the diffusion UNet (Ronneberger et al., 2015) backbone, leading to longer training pipelines and modifications to the generated image layout. RoSteALS (Bui et al., 2023) and work (Meng et al., 2024) imprint watermarks into the latent space of the VAE (Kingma & Welling, 2014), but face challenges with unstable training, requiring either multi-stage training processes or precise hyperparameter adjustments. FSW (Xiong et al., 2023), StableSignature (Fernandez et al., 2023), and WOUAF (Kim et al., 2024) also inject watermarks into the VAE feature space but necessitate modifications to the VAE decoder, often degrading the quality of the generated images (Ci et al., 2024a). Besides, these methods generally require collecting extensive external data for training and remain prone to artifacts on watermarked images. In contrast, our method requires no external data and features a plugin-based design, offering convenience while preserving high image quality.

## 3 BACKGROUND

**Notations and Definitions.** Let $(\mathcal{X}, \rho)$ be a metric space, where $\rho(\mathbf{x}, \mathbf{y})$ is a distance function for two instances $\mathbf{x}$ and $\mathbf{y}$ in the space $\mathcal{X}$. Similarly, let $(\mathcal{Y}, \rho')$ be another metric space, and $K > 0$ be a real number. A function $f : \mathcal{X} \to \mathcal{Y}$ is termed $K$-Lipschitz continuous if for any $\mathbf{x}, \mathbf{y} \in \mathcal{X}$, the following inequality holds:

$$\rho'(f(\mathbf{x}), f(\mathbf{y})) \leq K\rho(\mathbf{x}, \mathbf{y}). \tag{1}$$

Note that the smallest $K$ that satisfies Eq. (1) is known as the *Lipschitz constant* or *Lipschitz norm* of $f$, denoted by $\|f\|_{\text{Lip}}$. Next, we recall the concept of a *push-forward measure*. Consider a probability distribution $\mu$ on the space $\mathcal{X}$, then the push-forward measure, denoted $f\sharp\mu$, is a

probability distribution on $\mathcal{Y}$ defined by $f \sharp \mu(A) = \mu(f^{-1}(A))$ for any measurable set $A \subseteq \mathcal{Y}$. In essence, to sample from $f \sharp \mu$, one first samples $\mathbf{x}$ from $\mu$ and then sets $\mathbf{y} = f(\mathbf{x})$.

**Previous Training-based Methods for Diffusion-native Watermarking in Brief.** In previous works (Fernandez et al., 2023; Xiong et al., 2023; Bui et al., 2023; Meng et al., 2024; Ci et al., 2024a), the pipeline for training LDM to achieve watermarking can be formalized as a message embedding stage followed by a message extracting stage:

$$\begin{aligned} \text{Embedding}: \quad & \mathcal{X} \times \mathcal{M} \to \mathcal{X}, \quad \mathrm{E}_m\left(\mathbf{I}, \mathbf{m}\right) = \mathrm{D}_m\left(\mathrm{E}\left(\mathbf{I}\right), \mathbf{m}\right) = \mathbf{I}_w, \\ \text{Extracting}: \quad & \mathcal{X} \to \mathcal{M}, \quad \mathrm{T}_m(\phi(\mathbf{I}_w)) = \mathbf{m}', \end{aligned} \quad (2)$$

where $\mathcal{X}$ and $\mathcal{M}$ represent the image space and the message space, respectively. The training image $\mathbf{I}$ and the watermarked image $\mathbf{I}_w$ are both elements of $\mathcal{X}$. The message to be embedded, denoted as $\mathbf{m}$, belongs to $\mathcal{M}$. The message encoder $\mathrm{E}_m$ comprises the VAE encoder $\mathrm{E}$ and a modified decoder $\mathrm{D}_m$. The decoder $\mathrm{D}$ in the original VAE is altered to $\mathrm{D}_m$ to embed the message $\mathbf{m}$. The message extractor $\mathrm{T}_m$ is used to extract the message $\mathbf{m}'$ from the attacked image $\phi(\mathbf{I}_w)$, where $\phi$ is a transformation function for attacking watermarked image. The objective of training can be termed as minimizing the following loss over the input image distribution $\mu_x$ on $\mathcal{X}$ and the message distribution $\mu_m$ on $\mathcal{M}$:

$$\mathop{\mathbb{E}}_{\mathbf{I} \sim \mu_x} \mathop{\mathbb{E}}_{\mathbf{m} \sim \mu_m} \left[\ell_m\left(\mathbf{m}, \mathbf{m}', \boldsymbol{\lambda}_m\right) + \ell_I\left(\mathbf{I}_o, \mathbf{I}_w, \boldsymbol{\lambda}_I\right)\right], \quad (3)$$

where $\mathbf{I}_o$ is the generated image from the original decoder $\mathrm{D}$, i.e., $\mathbf{I}_o = \mathrm{D}\left(\mathrm{E}(\mathbf{I})\right)$. $\ell_m$ is a function that measures the discrepancy between $\mathbf{m}'$ and $\mathbf{m}$, and $\ell_I$ measures the discrepancy between $\mathbf{I}_o$ and $\mathbf{I}_w$. $\boldsymbol{\lambda}_m$ and $\boldsymbol{\lambda}_I$ are weights related to $\ell_m$ and $\ell_I$, respectively. $\ell_m$ and $\ell_I$ are designed according to specific requirements and can be combinations of multiple functions. For example, $\ell_I$ can be a weighted sum of $L_2$ residual regularization and LPIPS perceptual loss (Zhang et al., 2018), i.e., $\ell_I\left(\mathbf{I}_o, \mathbf{I}_w, \boldsymbol{\lambda}_I\right) = \lambda_I^1 \mathrm{MSE}(\mathbf{I}_o, \mathbf{I}_w) + \lambda_I^2 \mathrm{LPIPS}(\mathbf{I}_o, \mathbf{I}_w)$.

**Wasserstein Metric.** The *p-th Wasserstein distance* between two probability measures $\mu$ and $\mu'$ is defined as:

$$W_p(\mu, \mu') = \left(\inf_{\gamma \in \Pi(\mu, \mu')} \int \rho(\mathbf{x}, \mathbf{y})^p d\gamma(\mathbf{x}, \mathbf{y})\right)^{1/p}, \quad (4)$$

where $\mu, \mu' \in \{\gamma : \int \rho(\mathbf{x}, \mathbf{y})^p d\gamma(\mathbf{x}) < \infty, \forall \mathbf{y} \in \mathcal{X}\}$ are two probability measures on $(\mathcal{X}, \rho)$ with finite $p$-th moment, and $\Pi(\mu, \mu')$ represents the set of all measures on $\mathcal{X} \times \mathcal{X}$ with marginals $\mu$ and $\mu'$. The Wasserstein metric is relevant in the context of optimal transport: $\gamma(\mathbf{x}, \mathbf{y})$ can be interpreted as a randomized policy for moving a unit quantity of material from a random location $\mathbf{x}$ to another location $\mathbf{y}$ while adhering to the marginal constraint $\mathbf{x} \sim \mu$ and $\mathbf{y} \sim \mu'$. If the cost of transporting a unit of material from $\mathbf{x} \in \mu$ to $\mathbf{y} \in \mu'$ is given by $\rho(\mathbf{x}, \mathbf{y})^p$, then $W_p(\mu, \mu')$ is the minimal expected transport cost. The Kantorovich-Rubinstein theorem (Villani, 2009) reveals that when $\mathcal{X}$ is separable, the dual representation of the *first Wasserstein distance* (*Earth-Mover distance*) can be expressed as an integral probability metric:

$$W_1(\mu, \mu') = \sup_{\|f\|_{\mathrm{Lip}} \leq 1} \mathbb{E}_{\mathbf{x} \sim \mu}[f(\mathbf{x})] - \mathbb{E}_{\mathbf{x} \sim \mu'}[f(\mathbf{x})], \quad (5)$$

where $\|f\|_{\mathrm{Lip}} \leq 1$ denotes the set $\{f \mid f : \mathcal{X} \to \mathbb{R}, \|f\|_{\mathrm{Lip}} \leq 1\}$. For simplicity, the term "Wasserstein distance" in the following text refers specifically to the first Wasserstein distance.

# 4 SELF-AUGMENTED TRAINING: A SIMPLE YET EFFECTIVE METHOD

In this section, we introduce Self-Augmentation Training (SAT), inspired by our observations of the discrepancies between the training and testing phases in previous methods. Specifically, during the testing phase, the pipeline involves the sequence: prompt $\to$ UNet $\to$ VAE decoder, whereas during the previous training phase, it follows: image $\to$ VAE encoder $\to$ VAE decoder. This inconsistency likely limits the generalization ability of the watermarking module. Formally, during the testing phase, the pipeline for the LDM to generate watermarked images is summarized as follows:

$$\begin{aligned} \text{Embedding}: \quad & \mathcal{P} \times \mathcal{E} \times \mathcal{M} \to \mathcal{X}, \quad \mathrm{G}_m\left(\mathbf{x}^{\text{prompt}}, \boldsymbol{\epsilon}, \mathbf{m}\right) = \mathrm{D}_m\left(\mathrm{U}\left(\mathbf{x}^{\text{prompt}}, \boldsymbol{\epsilon}\right), \mathbf{m}\right) = \mathbf{I}_w, \\ \text{Extracting}: \quad & \mathcal{X} \to \mathcal{M}, \quad \mathrm{T}_m(\phi(\mathbf{I}_w)) = \mathbf{m}', \end{aligned} \quad (6)$$

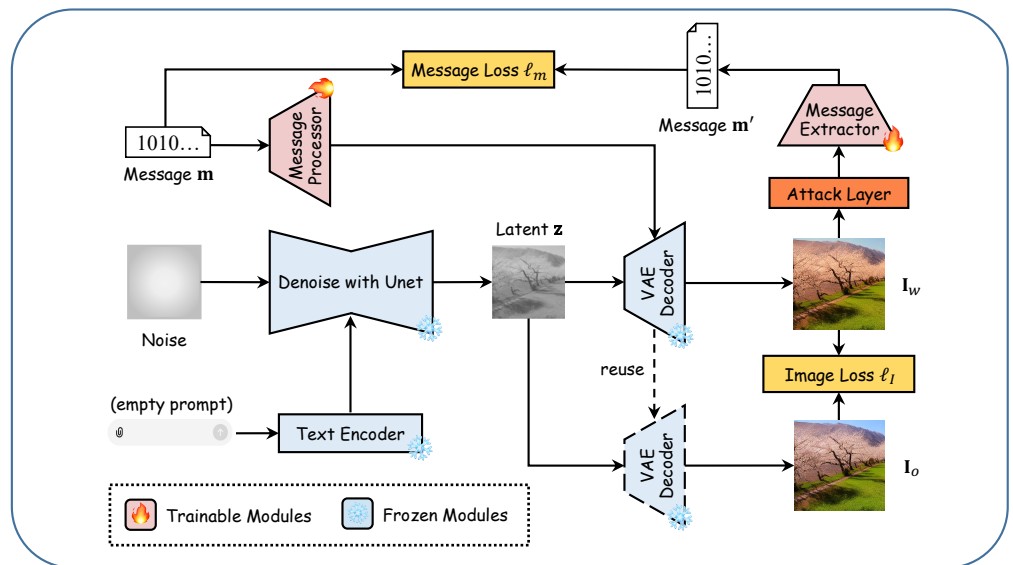

Figure 2: **The training pipeline of the proposed SAT-LDM.** The message processor is plugged onto the VAE decoder. Unlike conventional methods, SAT-LDM utilizes a self-augmented training mechanism that aligns the training and testing phases, thereby enhancing watermark effectiveness across diverse image styles without the need for external datasets.

where $\mathcal{P}$ and $\mathcal{E}$ represent the prompt space and noise space respectively, $\boldsymbol{\epsilon} \in \mathcal{E}$ is the noise sampled during the denoising process, and the image generation model G is modified to obtain the watermarked image generation model $G_m$. This model consists of a denoising process U and a modified-decoder $D_m$. Comparing Eq. (2) and Eq. (6), there is an inconsistency in the embedding stage. Hence, it is natural to align them, which leads to the following loss:

$$\mathbb{E}_{\mathbf{x}^{\text{prompt}} \sim \mu_p} \mathbb{E}_{\boldsymbol{\epsilon} \sim \mu_\epsilon} \mathbb{E}_{\mathbf{m} \sim \mu_m} \left[ \ell_m \left( \mathbf{m}, \mathbf{m}', \boldsymbol{\lambda}_m \right) + \ell_I \left( \mathbf{I}_o, \mathbf{I}_w, \boldsymbol{\lambda}_I \right) \right], \tag{7}$$

where $\mu_p$ and $\mu_\epsilon$ are distributions on $\mathcal{P}$ and $\mathcal{E}$, respectively. Nevertheless, the process of sampling $\mathbf{x}^{\text{prompt}}$ presents a substantial challenge due to the inherent uncertainty surrounding the true prompt distribution $\mu_p$. The actual distribution is not only unknown but also difficult to approximate accurately. Even though we can leverage prompts from publicly available datasets for training, these sources often carry inherent biases and fail to comprehensively represent the wide variability and diversity of prompts the model might encounter in real-world applications. As a result, we shift our focus to modeling the distribution derived from $\mathbf{x}^{\text{prompt}}$ and $\boldsymbol{\epsilon}$.

Toward this end, we introduce a formal definition that aids in conceptualizing the latent representations generated by LDM. Specifically, we define $\mathbf{z} = U\left(\mathbf{x}^{\text{prompt}}, \boldsymbol{\epsilon}\right)$ be the image latent representation and $\mu_z = U\sharp\left(\mu_p \times \mu_\epsilon\right)$ be the corresponding distribution. In other words, sampling $\mathbf{z} \sim \mu_z = U\sharp\left(\mu_p \times \mu_\epsilon\right)$ means first sampling $\mathbf{x}^{\text{prompt}} \sim \mu_p$ and $\boldsymbol{\epsilon} \sim \mu_\epsilon$, then setting $\mathbf{z} = U\left(\mathbf{x}^{\text{prompt}}, \boldsymbol{\epsilon}\right)$. Ideally, the influence of prompt can be averaged overall, i.e., $p\left(\mathbf{z} \mid \boldsymbol{\epsilon}\right) = \sum p\left(\mathbf{z} \mid \boldsymbol{\epsilon}, \mathbf{x}^{\text{prompt}}\right) p\left(\mathbf{x}^{\text{prompt}}\right)$, and then the distribution of latent representations generated by all prompts via conditional sampling (conditional generation distribution) is equivalent to the one without a specific prompt[1] (free generation distribution), i.e., $U\sharp\left(\mu_p \times \mu_\epsilon\right) = U\sharp\mu_\epsilon \Leftrightarrow G\sharp\left(\mu_p \times \mu_\epsilon\right) = G\sharp\mu_\epsilon$.

Actually, this equality may not hold in practical scenarios due to the model's parameters and training data limitations. Specifically, suboptimal training or insufficient pre-training data can lead to biases in how prompts influence generated samples. Nonetheless, this assumption provides a useful simplification for our analysis, and the impact of such discrepancies is often minimal in many applications. Moreover, from a technical perspective, we quantify and visualize the similarity between the conditional generation distribution and free generation distribution using the Wasserstein metric and t-SNE method in the Section 5.3. This empirical analysis provides further support for our theoretical claims,

---

[1]Typically, sampling without specific prompt is achieved by setting $\mathbf{x}^{\text{prompt}}$ to an empty string ""

reinforcing the robustness of our findings. Based on this analysis, the Eq. 7 can be simplified as:

$$\mathop{\mathbb{E}}_{\boldsymbol{\epsilon} \sim \mu_\epsilon} \mathop{\mathbb{E}}_{\mathbf{m} \sim \mu_m} \left[ \ell_m \left( \mathbf{m}, \mathbf{m}', \boldsymbol{\lambda}_m \right) + \ell_I \left( \mathbf{I}_o, \mathbf{I}_w, \boldsymbol{\lambda}_I \right) \right], \tag{8}$$

where $\mathbf{I}_w = \mathrm{G}_m \left( \text{``''}, \boldsymbol{\epsilon}, \mathbf{m} \right)$ represents the watermarked image generated without a specific prompt.

## 4.1 THEORETICAL ANALYSIS

We next provide an analysis of the generalization bound of the self-augmented training method, comparing it to previous approaches to highlight the advantages of our approach. Given a data-generating distribution $\mu_x$ on the Euclidean observation space $\mathcal{X}$, a probability measure $\mu_z$ on the latent space $\mathcal{Z} = \mathbb{R}^{d_\mathcal{Z}}$, a probability measure $\mu_\epsilon$ on the noise space $\mathcal{E}$, and a hypothesis class $\mathcal{H} = \{(\mathrm{D}_m, \mathrm{T}_m) \mid \mathrm{D}_m : \mathcal{Z} \times \mathcal{M} \to \mathcal{X}, \mathrm{T}_m : \mathcal{X} \to \mathcal{M}\}$, we introduce a unified loss function:

$$\ell(h, \mathbf{z}, \mathbf{m}) = \ell_m \left( \mathrm{T}_m(\mathrm{D}_m(\mathbf{z}, \mathbf{m})), \mathbf{m}, \boldsymbol{\lambda}_m \right) + \ell_I \left( \mathrm{D}_m(\mathbf{z}, \mathbf{m}), \mathrm{D}(\mathbf{z}), \boldsymbol{\lambda}_I \right), \tag{9}$$

where $h \in \mathcal{H}$, and $(\mathbf{z}, \mathbf{m}) \sim \mu_z \times \mu_m$: $\mu_z = \mathrm{E} \sharp \mu_x$ for the previous training method; $\mu_z = \mathrm{U} \sharp \mu_\epsilon$ for the proposed training method. We begin by proving an intermediate lemma.

**Lemma 1.** *Let $(\mathcal{Z}, \rho_\mathcal{Z})$ and $(\mathcal{M}, \rho_\mathcal{M})$ be two metric spaces, $\mu_z$ and $\mu_t$ be two probability measures on $\mathcal{Z}$, and $\mu_m$ be a probability measure on $\mathcal{M}$. For $(\mathbf{z}, \mathbf{m}), (\mathbf{z}', \mathbf{m}') \in \mathcal{Z} \times \mathcal{M}$, the distance function is defined as $\rho_{\mathcal{Z}, \mathcal{M}}((\mathbf{z}, \mathbf{m}), (\mathbf{z}', \mathbf{m}')) = \rho_\mathcal{Z}(\mathbf{z}, \mathbf{z}') + \rho_\mathcal{M}(\mathbf{m}, \mathbf{m}')$. Then we can obtain:*

$$W_1(\mu_t \times \mu_m, \mu_z \times \mu_m) = W_1(\mu_t, \mu_z). \tag{10}$$

Then we introduce the Wasserstein distance to link the training error and the testing error.

**Theorem 1.** *Under the definitions of Lemma 1, consider a hypothesis class $\mathcal{H}$, a loss function $\ell : \mathcal{H} \times \mathcal{Z} \times \mathcal{M} \to \mathbb{R}$ and real numbers $\delta \in (0, 1)$. Assume that for hypotheses $h \in \mathcal{H}$, the loss function $\ell$ is $K$-Lipschitz continuous for some $K$ w.r.t. $(\mathbf{z}, \mathbf{m}) \in \mathcal{Z} \times \mathcal{M}$, and is bounded within an interval $G$: $G = max(\ell)$ - $min(\ell)$. With probability at least $1 - \delta$ over the random draw of $\{(\mathbf{z}_1, \mathbf{m}_1), \cdots, (\mathbf{z}_n, \mathbf{m}_n)\} \sim (\mu_z \times \mu_m)^{\otimes n}$, for every hypothesis $h \in \mathcal{H}$:*

(1) Empirical risk

(2) Deviation term

$$\mathop{\mathbb{E}}_{(\mathbf{z}, \mathbf{m}) \sim \mu_t \times \mu_m} \left[ \ell(h, \mathbf{z}, \mathbf{m}) \right] \leq \boxed{\tfrac{1}{n} \sum_{i=1}^n \ell \left( h, \mathbf{z}_i, \mathbf{m}_i \right)} + \boxed{\sqrt{\tfrac{G^2 \log(1/\delta)}{2n}}} + \boxed{K W_1(\mu_t, \mu_z)}. \tag{11}$$

(3) Distributional difference

Theorem 1 bounds the combined expected loss of the watermarked image generator $\mathrm{G}_m$ and the message extractor $\mathrm{T}_m$. Upon receiving previously unseen image latent representation-message pairs $(\mathbf{z}, \mathbf{m}) \sim \mu_t \times \mu_m$, $\mathrm{G}_m$ generates the watermarked image, and $\mathrm{T}_m$ extracts it. We aim to minimize this generalization bound as much as possible. It consists of three components: (1) the empirical risk reflects the model's performance on the training data and is generally minimized through proper optimization. (2) the deviation term quantifies the difference between empirical and expected risks, diminishing with an increased sample size. (3) The most critical factor is the distributional difference, which is tied to the Wasserstein distance $W_1(\mu_t, \mu_z)$. In our context, the regularization stabilizes the Lipschitz constant $K$, so the distributional discrepancy dominates.

The detailed proofs are provided in Appendix A.1 and Appendix A.2. The theorem assumes that the loss function $\ell$ is $K$-Lipschitz continuous and bounded. These assumptions are applicable to many machine learning models, which are commonly satisfied by standard loss functions (e.g., mean squared error, cross-entropy) (Bousquet & Elisseeff, 2002; Zhang et al., 2021) and neural networks, which employ Lipschitz continuous activation functions (Bartlett et al., 2017; Ledoux, 2001) and regularization methods (Bengio et al., 2017; Srivastava et al., 2014). See Appendix G for more discussion on Lipschitz continuity of the loss function.

**Remark.** In conclusion, when the model's structure, loss function, and sample size are fixed and the model is well-trained, the generalization bound is primarily influenced by $W_1(\mu_t, \mu_z)$. This finding implies that selecting an appropriate training distribution to minimize $W_1(\mu_t, \mu_z)$ can effectively reduce the model's generalization error. In our task, the test distribution is defined as

$\mu_t = \mathrm{U} \sharp (\mu_p \times \mu_\epsilon)$, where $\mu_p$ represents the probability distribution over the prompt space. The watermarking module is designed to handle a wide variety of styles, meaning $\mu_p$ should cover any sentence or word. Referring to Eq. (9), previous methods use an external data generation distribution, $\mu_z = \mathrm{E} \sharp \mu_x$, while our approach employs a free generation distribution, $\mu_z = \mathrm{U} \sharp \mu_\epsilon$. Since $\mu_z = \mathrm{E} \sharp \mu_x$ depends on external data, it is difficult to regulate its behavior to approximate the test distribution closely. Given that $\mathrm{U} \sharp (\mu_p \times \mu_\epsilon) = \mathrm{U} \sharp \mu_\epsilon$, our free generation distribution aligns more closely with the test distribution, leading to a smaller $W_1$ value and reduced generalization error, as corroborated by the experiments discussed in Section 5.3 and Appendix E.3.

## 4.2 Implementation Details

**Architectures.** We employ the FSW (Xiong et al., 2023) model structure, which is one of the latest training-based invisible watermarking methods. It jointly optimizes the parameters of the modified-decoder $\mathrm{D}_m$ and message-extractor $\mathrm{T}_m$ to embed $l$-bit messages into images, ensuring robustness to transformations applied during training. $\mathrm{D}_m$ introduces a message processor to process the input message $\mathbf{m}$ into a message matrix, which is then fused with the outputs of selected intermediate layers (namely, the input convolutional layers, intermediate blocks, and the first four upsampling modules) during the image generation process to create watermarked images. The message processor mainly consists of 5 fully connected (FC) layers and 6 convolutional layers, and the message extractor mainly comprises 7 convolutional layers and 3 FC layers. For further details, we refer the reader to the original paper (Xiong et al., 2023). We implement two modifications to the model structure. (1) We aim to ensure the entity which manages the model can easily replace the watermarked model with a non-watermarked version without disrupting the user experience. To achieve this, we preserve the parameters of the original VAE decoder and replicate the structure of selected intermediate layers to integrate them into the message processor for training. This method retains the original message fusion process while maintaining the integrity of the original VAE decoder's parameters. (2) We observe that $\mathrm{T}_m$ has limited robustness against perspective changes, even with adversarial training added during training. To address this, we add a spatial transformer network (Jaderberg et al., 2015) in front of $\mathrm{T}_m$ to improve robustness to slight perspective changes introduced when images are printed and photographed.

**Loss Function and Training strategy.** Then we introduce the design of loss in Eq. (9). The image loss is defined as the combination of following functions:

$$\ell_I \left( \mathbf{I}_o, \mathbf{I}_w, \boldsymbol{\lambda}_I \right) = \lambda_I^1 \, \mathrm{MSE}(\mathbf{I}_o, \mathbf{I}_w) + \lambda_I^2 \, \mathrm{LPIPS}(\mathbf{I}_o, \mathbf{I}_w) + \lambda_I^3 \, \mathrm{BAL}(\mathbf{I}_o, \mathbf{I}_w),$$

$$\mathrm{BAL} \left( \mathbf{I}, \mathbf{I}' \right) = \frac{1}{c \cdot h \cdot w} \sum_{i=1}^{c} \sum_{j=1}^{h} \sum_{k=1}^{w} \frac{\left| \mathbf{I}_{(i,j,k)} - \mathbf{I}'_{(i,j,k)} \right|}{\mathbf{I}_{(i,j,k)} + 1}, \tag{12}$$

where BAL is used to balance the impact of the watermark on each pixel, preventing excessive modification of pixels with smaller values (Xiong et al., 2023). Here, $c$, $h$, and $w$ represent the image's channels, height, and width, respectively. The message loss is simply designed as:

$$\ell_m \left( \mathbf{m}, \mathbf{m}', \boldsymbol{\lambda}_m \right) = \lambda_m^1 \, \mathrm{MSE}(\mathbf{m}, \mathbf{m}'). \tag{13}$$

During training, we use the AdamW optimizer (Loshchilov & Hutter, 2017) with a learning rate of $2 \times 10^{-5}$. In terms of watermark robustness, we also align with FSW to ensure fairness and use an attack layer with seven types of watermark attacks, as detailed in Section B. At each training step, the watermarked image $\mathbf{I}_w$ undergoes the attack layer with a random attack intensity, then is processed by the message-extractor $\mathrm{T}_m$. Note that the attack intensity is scaled by a decay coefficient $\gamma_\phi$, which gradually increases from 0 to 1 to assist in convergence. Figure 2 shows the training pipeline and Algorithm 1 describes the training procedure.

# 5 Experiments

## 5.1 Experimental Setup

**SD models.** In this paper, we focus on text-to-image LDM, and thus we choose the SD (Rombach et al., 2022) provided by huggingface. We use the commonly used version v1.5 of SD [2] to evaluate the

---

[2]https://huggingface.co/stable-diffusion-v1-5/stable-diffusion-v1-5.

Table 1: **The overall performance on quality of the watermarked images and robustness of the watermarking methods.** The numbers 1 to 7 represent Gaussian blur, Gaussian noise, brightness, contrast, desaturation, perspective warp, and JPEG attacks, respectively.

(a) Comparison with several competitive watermarking methods.

| Methods | PSNR↑ | SSIM↑ | FID↓ | Bit accuracy↑ | | | | | | | | |
|---|---|---|---|---|---|---|---|---|---|---|---|---|
| | | | | None | 1 | 2 | 3 | 4 | 5 | 6 | 7 | Adv. |
| DwtDctSvd | 37.94 | **0.984** | 7.04 | 0.999 | 0.982 | 0.965 | 0.994 | 0.608 | 0.527 | 0.606 | 0.959 | 0.806 |
| HiDDeN | 25.56 | 0.841 | 57.94 | 0.966 | 0.888 | 0.775 | 0.948 | 0.940 | 0.873 | 0.776 | 0.690 | 0.842 |
| StegaStamp | 28.02 | 0.921 | 29.25 | 0.999 | **0.999** | **0.997** | 0.996 | 0.996 | **0.998** | **0.990** | **0.998** | **0.996** |
| Stable Signature | 29.17 | 0.946 | 9.49 | 0.958 | 0.813 | 0.932 | 0.951 | 0.941 | 0.920 | 0.930 | 0.883 | 0.910 |
| FSW | 27.32 | 0.891 | 18.92 | 0.997 | 0.996 | 0.996 | 0.995 | 0.996 | 0.961 | 0.568 | 0.987 | 0.928 |
| SAT-LDM | **40.58** | 0.983 | **2.40** | **1.000** | 0.981 | 0.995 | **0.998** | **0.998** | 0.994 | 0.980 | 0.968 | 0.988 |

(b) Comparison between training distributions (External v.s. Free) and the Wasserstein distance ($W_1$) between training and testing distributions.

| Training distributions | PSNR↑ | SSIM↑ | FID↓ | Bit accuracy↑ | | | | | | | | | $W_1$ ↓ |
|---|---|---|---|---|---|---|---|---|---|---|---|---|---|
| | | | | None | 1 | 2 | 3 | 4 | 5 | 6 | 7 | Adv. | |
| External | 37.46 | 0.987 | 3.50 | 1.000 | 0.974 | 0.999 | 1.000 | 1.000 | 1.000 | 0.929 | 0.975 | 0.982 | 911.4 |
| Free | 40.58 | 0.983 | 2.40 | 1.000 | 0.981 | 0.995 | 0.998 | 0.998 | 0.994 | 0.980 | 0.968 | 0.988 | 504.4 |

proposed methods as well as baseline methods. The generated images have a size of $512 \times 512$, with a latent space dimension of $4 \times 64 \times 64$. During both training and testing, we utilize DDPM (Ho et al., 2020) sampling with 30 steps. The sample size for free generation distribution is 30K. In the testing phase, we aim to use prompts that cover a variety of styles and complexities to comprehensively evaluate the generalization performance of the watermark. Toward this end, we use GPT-4 to generate 10 diverse categories of image prompts, as depicted in Appendix C. For each category, 100 prompts with varying styles are generated, totaling 1K prompts (AI-Generated Prompts). Then, SD generates images from these prompts as test data, with a guidance scale of 7.5.

**Baselines.** In the main experiment, the number of message bits $l$ is set to 100. For the baseline, we select post-hoc watermarking methods (including DwtDctSvd (Cox et al., 2007) used by SD officially, HiDDeN (Zhu et al., 2018), and StegaStamp (Tancik et al., 2020)) as well as a training based method (Stable Signature Fernandez et al. (2023) and FSW (Xiong et al., 2023)).

**Robustness Evaluation.** To emulate practical scenarios, the watermarked images are first saved in PNG format. Subsequently, they undergo a sequence of operations: reading, applying an attack, re-saving, and finally, extracting the embedded message. It is important to note that the attack intensity decay coefficient, denoted as $\gamma_\phi$, is fixed at 1.

**Evaluation Metrics.** Referring to previous studies (Xiong et al., 2023; Fernandez et al., 2023), our metrics are divided into two different aspects: the quality of the watermarked image and the robustness of the watermark. For the quality of the watermarked images, we use Peak Signal-to-Noise Ratio (PSNR), Structural Similarity Index (SSIM) (Wang et al., 2004), and Fréchet Inception Distance (FID) (Heusel et al., 2017) to measure the pixel-level and feature-level differences between the watermarked and non-watermarked generated images. Specifically, $\text{PSNR}(\mathbf{I}, \mathbf{I}') = -10 \cdot \log_{10}(\text{MSE}(\mathbf{I}, \mathbf{I}'))$. Bit accuracy (percentage of correctly decoded bits) is used to evaluate the robustness of the watermark. Since LDM is computation-intensive and we observe the results fluctuate marginally, the experimental section presents the outcomes of a single experiment.

## 5.2 MAIN RESULTS

We compare the performance of the proposed method with the baseline methods. All methods use a message bit number of 100, except for HiDDeN and Stable Signature, where we use the open-source

reproduced model[3] and the pre-trained model[4] with a message bit number of 48, respectively. For StegaStamp and FSW, we utilize the pre-trained models provided by the original papers. Due to the limitations of the specially designed networks, the image size for StegaStamp is set to $400 \times 400$.

We test 1K generated images for each method. Table 1a lists the comparisons of PSNR, SSIM, FID, and bit accuracy. The proposed SAT-LDM demonstrates strong image quality, with PSNR and FID significantly better than other baselines, and SSIM comparable to the best baseline in this metric. Specifically, the FID score of 2.40 highlights the model's capability to generate watermarked images that are nearly indistinguishable from non-watermarked ones, outperforming even the closest competitor by a margin of over 50%. The proposed watermarking method demonstrates strong robustness, achieving a bit accuracy of over 96%. Moreover, we also evaluate on other public datasets in the same way. Figure 1 shows that SAT-LDM consistently performs well across different datasets, which further demonstrates its superior generalization across various image styles. Such generalization makes SAT-LDM more suitable for real-world applications where diverse image styles and robustness against attacks are crucial.

## 5.3 EXPERIMENTAL ANALYSIS

Here, we conduct comprehensive ablation studies to show that (1) free generation distributions significantly reduce generalization error, (2) moderate training sample sizes yield optimal performance, and (3) the system exhibits strong robustness to varying message lengths, sampling methods, guidance scales, and inference steps. These insights validate the design choices in SAT-LDM and highlight its flexibility and effectiveness. See Appendix E for additional experimental analysis. Unless stated otherwise, the experimental setup follows the description provided in Section 5.1.

**Training Distributions.** We assess the impact of the training distribution $\mu_z$ on generalization performance by retraining our model using 30K images from the LAION-400M dataset (Schuhmann et al., 2021), which is included in the LAION-5B dataset used to train the official SD model (Schuhmann et al., 2022). As shown in Table 1b, the model trained with external data shows lower performance in PSNR, FID, and robustness to perspective warp. To compare distributions, we sampled 1K instances from both the external and free generation distributions, using them as proxies for their respective training distributions. These are compared to the test distribution generated from the test data. Figure 3 visualizes these distributions via t-SNE, where the free generation distribution closely aligns with the test distribution, while the external data generation distribution diverges. We also calculated the Wasserstein distances between the training and test distributions, as shown in Table 1b. These results strongly support our theoretical framework, highlighting the superior alignment of free generation with the test distribution. By minimizing distributional discrepancy, our method improves generalization, providing a more robust foundation for practical applications.

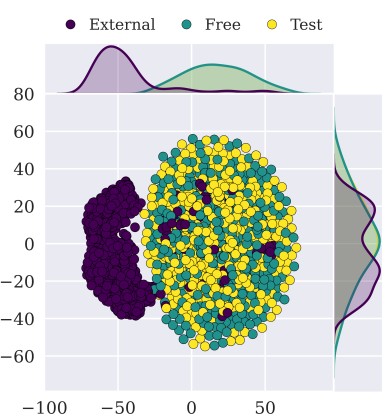

Figure 3: **The t-SNE visualization of proxies for external data, free, and conditional/test distributions.** The free generation distribution closely aligns with the test distribution, while the external data generation distribution diverges.

**Number of Training Samples.** As shown in Table 2, with the increase in the number of samples, the quality of the watermarked images remains relatively stable, but the robustness first increases and then decreases. We posit the reason is that the proposed method is designed to generate watermarked images from any given prompt, including meaningless ones, such as gibberish. In contrast, the prompts we use here are all meaningful. Hence, as the number of training samples increases, the training distribution begins to align with and then deviates from the test distribution. Furthermore,

---

[3]https://github.com/ando-khachatryan/HiDDeN.

[4]https://github.com/facebookresearch/stable_signature.

an excessively large training set introduces additional computational overhead. Based on these observations, we set the number of training samples to 30K to strike a balance between robustness and efficiency, which is much smaller than many previous methods (Bui et al., 2023; Fernandez et al., 2023; Xiong et al., 2023; Ci et al., 2024a).

**Number of Message Bits.** Table 2 compares five separately trained models with different message lengths. They all perform well in terms of SSIM, PSNR, and FID, but the robustness decreases when the length is longer, though it still remains above 90%. This is because larger messages are more difficult to embed and extract. Fortunately, given the excellent quality of the watermarked images produced by SAT-LDM, robustness can be further improved by sacrificing some image quality (Fernandez et al., 2023), which can be achieved by adjusting $\lambda_I$ and $\lambda_m$.

**Sampling Methods.** To verify generalization ability, we select four representative sampling methods, i.e., DDPM (Ho et al., 2020), DDIM (Song et al., 2021), LMS[5] and Euler (Karras et al., 2022). As shown in Table 2, they all exhibit excellent performance, with a bit accuracy of approximately 97% in the presence of attack.

**Guidance Scales.** The guidance scale adjusts the balance between following the prompt and allowing creative freedom, and it may influence the shift from the free generation distribution. In SD, the typical range for guidance scales is between 5 and 15, so we extend our experiments from 2 to 18. As shown in Table 2, increasing the guidance scale leads to a

Table 2: **Performance ablations of SAT-LDM** with different number of training samples, number of message bits, sampling methods, guidance scales and inference steps.

| | | PSNR↑ | SSIM↑ | FID↓ | Bit accuracy↑ | |
| | | | | | None | Adv. |
|---|---|---|---|---|---|---|
| #Training sample | 15K | 41.88 | 0.994 | 2.62 | 1.000 | 0.972 |
| | 30K | 40.58 | 0.983 | 2.40 | 1.000 | 0.988 |
| | 60K | 40.18 | 0.980 | 2.10 | 0.999 | 0.982 |
| | 120K | 41.54 | 0.994 | 2.70 | 1.000 | 0.963 |
| #Message bit | 48 | 43.92 | 0.996 | 2.21 | 1.000 | 0.979 |
| | 64 | 42.51 | 0.994 | 2.46 | 1.000 | 0.981 |
| | 100 | 40.58 | 0.983 | 2.40 | 1.000 | 0.988 |
| | 128 | 41.17 | 0.993 | 3.28 | 1.000 | 0.977 |
| | 200 | 40.51 | 0.993 | 3.25 | 0.999 | 0.970 |
| Sampling methods | DDPM | 40.58 | 0.983 | 2.40 | 1.000 | 0.988 |
| | DDIM | 39.93 | 0.986 | 2.35 | 1.000 | 0.989 |
| | LMS | 40.00 | 0.986 | 2.23 | 1.000 | 0.986 |
| | Euler | 40.14 | 0.984 | 2.41 | 1.000 | 0.987 |
| Guidance scales | 2 | 40.75 | 0.984 | 3.18 | 1.000 | 0.987 |
| | 6 | 40.62 | 0.983 | 2.60 | 1.000 | 0.994 |
| | 10 | 40.51 | 0.982 | 2.29 | 1.000 | 0.989 |
| | 14 | 40.44 | 0.982 | 2.32 | 1.000 | 0.985 |
| | 18 | 40.35 | 0.982 | 2.32 | 0.999 | 0.980 |
| Inference steps | 10 | 40.97 | 0.979 | 2.98 | 1.000 | 0.992 |
| | 30 | 40.58 | 0.983 | 2.40 | 1.000 | 0.988 |
| | 50 | 40.52 | 0.983 | 2.38 | 0.999 | 0.985 |
| | 100 | 40.49 | 0.984 | 2.29 | 1.000 | 0.984 |

slight decrease in the performance of the proposed method. This occurs because a higher guidance scale amplifies the influence of prompt and causes the test distribution to deviate from the free generation distribution in training, thus increasing the generalization error. Nonetheless, even raising the guidance scale to the rarely used value 18, it has a very weak degradation of performance, so our method is sufficiently robust to the guidance scale.

**Inference Steps.** In practice, the number of inference steps is often unknown, which introduces a mismatch with the free generation setup during training. As shown in Table 2, this mismatch causes minimal degradation in watermarked performance. Given the efficiency of modern samplers, the number of inference steps typically does not exceed 50. Therefore, we fix the number of inference steps to 30 for better efficiency in our experiments.

## 6 CONCLUSION AND FUTURE WORK

In this work, we propose SAT-LDM, an enhanced image watermarking method for latent diffusion model with self-augment training. Compared to existing methods, SAT-LDM offers effectiveness and convenience by utilizing the free generation distribution for training. This approach does not alter the diffusion process, ensuring compatibility with most LDM-based generative models. We also provide a theoretical analysis of the generalization error to consolidate our proposed method. Extensive experiments validate its superior performance, particularly in the quality of watermarked images. In future work, we will explore the application of dataset distillation (Wang et al., 2018) to further minimize the need for training samples, potentially eliminating that requirement altogether. Our prompt design may still harbor biases, necessitating future ablation studies to evaluate their impact comprehensively..

---

[5]https://en.wikipedia.org/wiki/Linear_multistep_method.

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

## A MATHEMATICAL PROOF

### A.1 PROOF OF LEMMA 1

This section presents the proof of Lemma 1, which establishes the equivalence of the Wasserstein distance for combined distributions.

**Lemma 1.** *Let $(\mathcal{Z}, \rho_{\mathcal{Z}})$ and $(\mathcal{M}, \rho_{\mathcal{M}})$ be two metric spaces, $\mu_z$ and $\mu_t$ be two probability measures on $\mathcal{Z}$, and $\mu_m$ be a probability measure on $\mathcal{M}$. For $(\mathbf{z}, \mathbf{m}), (\mathbf{z}', \mathbf{m}') \in \mathcal{Z} \times \mathcal{M}$, the distance function is defined as $\rho_{\mathcal{Z},\mathcal{M}}((\mathbf{z}, \mathbf{m}), (\mathbf{z}', \mathbf{m}')) = \rho_{\mathcal{Z}}(\mathbf{z}, \mathbf{z}') + \rho_{\mathcal{M}}(\mathbf{m}, \mathbf{m}')$. Then we can obtain:*

$$W_1(\mu_t \times \mu_m, \mu_z \times \mu_m) = W_1(\mu_t, \mu_z).$$

*Proof.* The Wasserstein distance $W_1(\mu_t \times \mu_m, \mu_z \times \mu_m)$ is defined as:

$$W_1(\mu_t \times \mu_m, \mu_z \times \mu_m) = \inf_{\gamma \in \Pi(\mu_t \times \mu_m, \mu_z \times \mu_m)} \int \rho_{\mathcal{Z},\mathcal{M}}((\mathbf{z}, \mathbf{m}), (\mathbf{z}', \mathbf{m}')) d\gamma((\mathbf{z}, \mathbf{m}), (\mathbf{z}', \mathbf{m}')),$$

where $\Pi(\mu_t \times \mu_m, \mu_z \times \mu_m)$ is the set of all couplings of $\mu_t \times \mu_m$ and $\mu_z \times \mu_m$. Since the distance function $\rho_{\mathcal{Z},\mathcal{M}}((\mathbf{z}, \mathbf{m}), (\mathbf{z}', \mathbf{m}')) = \rho_{\mathcal{Z}}(\mathbf{z}, \mathbf{z}') + \rho_{\mathcal{M}}(\mathbf{m}, \mathbf{m}')$, we can separate the integral to:

$$\inf_{\gamma \in \Pi(\mu_t \times \mu_m, \mu_z \times \mu_m)} \int (\rho_{\mathcal{Z}}(\mathbf{z}, \mathbf{z}') + \rho_{\mathcal{M}}(\mathbf{m}, \mathbf{m}')) \, d\gamma((\mathbf{z}, \mathbf{m}), (\mathbf{z}', \mathbf{m}')).$$

Given the independence between $\mu_z$ and $\mu_m$, we can divide the coupling $\gamma$ into two independent parts: one for $(\mathbf{z}, \mathbf{z}')$ and another for $(\mathbf{m}, \mathbf{m}')$. Thus:

$$\gamma((\mathbf{z}, \mathbf{m}), (\mathbf{z}', \mathbf{m}')) = \gamma_{\mathcal{Z}}(\mathbf{z}, \mathbf{z}') \cdot \gamma_{\mathcal{M}}(\mathbf{m}, \mathbf{m}')$$

Using this decomposition, we obtain:

$$\inf_{\gamma \in \Pi(\mu_t \times \mu_m, \mu_z \times \mu_m)} \int (\rho_{\mathcal{Z}}(\mathbf{z}, \mathbf{z}') + \rho_{\mathcal{M}}(\mathbf{m}, \mathbf{m}')) \, d\gamma((\mathbf{z}, \mathbf{m}), (\mathbf{z}', \mathbf{m}'))$$

$$= \inf_{\gamma \in \Pi(\mu_t, \mu_z)} \int \rho_{\mathcal{Z}}(\mathbf{z}, \mathbf{z}') \, d\gamma(\mathbf{z}, \mathbf{z}') + \inf_{\gamma \in \Pi(\mu_m, \mu_m)} \int \rho_{\mathcal{M}}(\mathbf{m}, \mathbf{m}') \, d\gamma(\mathbf{m}, \mathbf{m}')$$

$$= W_1(\mu_t, \mu_z) + W_1(\mu_m, \mu_m).$$

Note that the Wasserstein distance between identical distributions is zero (i.e., $W_1(\mu_m, \mu_m) = 0$), hence the second term vanishes:

$$W_1(\mu_t \times \mu_m, \mu_z \times \mu_m) = W_1(\mu_t, \mu_z).$$

$\square$

### A.2 PROOF OF THEORY 1

This section provides a detailed proof of Theorem 1, which quantifies the generalization bound for the proposed SAT-LDM method. The central result shows that the Wasserstein distance between the training and test distributions directly influences the generalization error, underscoring its importance in minimizing this error.

**Theorem 2.** *Under the definitions of Lemma 1, consider a hypothesis class $\mathcal{H}$, a loss function $\ell : \mathcal{H} \times \mathcal{Z} \times \mathcal{M} \to \mathbb{R}$ and real numbers $\delta \in (0, 1)$. Assume that for hypotheses $h \in \mathcal{H}$, the loss function $\ell$ is $K$-Lipschitz continuous for some $K$ w.r.t. $(\mathbf{z}, \mathbf{m}) \in \mathcal{Z} \times \mathcal{M}$, and is bounded within an interval $G$: $G = max(\ell)$ - $min(\ell)$. With probability at least $1 - \delta$ over the random draw of $\{(\mathbf{z}_1, \mathbf{m}_1), \cdots, (\mathbf{z}_n, \mathbf{m}_n)\} \sim (\mu_z \times \mu_m)^{\otimes n}$, for every hypothesis $h \in \mathcal{H}$:*

$$\mathbb{E}_{(\mathbf{z}, \mathbf{m}) \sim \mu_t \times \mu_m} [\ell(h, \mathbf{z}, \mathbf{m})] \leq \frac{1}{n} \sum_{i=1}^{n} \ell(h, \mathbf{z}_i, \mathbf{m}_i) + \sqrt{\frac{G^2 \log(1/\delta)}{2n}} + K W_1(\mu_t, \mu_z).$$

*Proof.* By the definition of the $K$-Lipschitz continuity of $\ell$, we have:

$$\mathbb{E}_{(\mathbf{z},\mathbf{m})\sim\mu_t\times\mu_m}[\ell(h,\mathbf{z},\mathbf{m})] - \mathbb{E}_{(\mathbf{z},\mathbf{m})\sim\mu_z\times\mu_m}[\ell(h,\mathbf{z},\mathbf{m})]$$

$$\leq \sup_{\|f\|_{\text{Lip}}\leq K}\left(\mathbb{E}_{\mathbf{x}\sim\mu_t\times\mu_m}[f(\mathbf{x})] - \mathbb{E}_{\mathbf{x}\sim\mu_z\times\mu_m}[f(\mathbf{x})]\right).$$

Using the Kantorovich-Rubinstein duality theorem mentioned in Section 3, this expression simplifies to:

$$\sup_{\|f\|_{\text{Lip}}\leq K}\left(\mathbb{E}_{\mathbf{x}\sim\mu_t\times\mu_m}[f(\mathbf{x})] - \mathbb{E}_{\mathbf{x}\sim\mu_z\times\mu_m}[f(\mathbf{x})]\right) = KW_1(\mu_t\times\mu_m,\mu_z\times\mu_m).$$

From Lemma 1, we know that:

$$W_1(\mu_t\times\mu_m,\mu_z\times\mu_m) = W_1(\mu_t,\mu_z).$$

Thus, we can bound the expected loss difference by:

$$\mathbb{E}_{(\mathbf{z},\mathbf{m})\sim\mu_t\times\mu_m}[\ell(h,\mathbf{z},\mathbf{m})] - \mathbb{E}_{(\mathbf{z},\mathbf{m})\sim\mu_z\times\mu_m}[\ell(h,\mathbf{z},\mathbf{m})] \leq KW_1(\mu_t,\mu_z).$$

Next, we apply Hoeffding's inequality to bound the empirical loss difference:

$$\Pr\left(\mathbb{E}_{(\mathbf{z},\mathbf{m})\sim\mu_z\times\mu_m}\ell(h,\mathbf{z},\mathbf{m}) - \frac{1}{n}\sum_{i=1}^{n}\ell(h,\mathbf{z}_i,\mathbf{m}_i) \leq \epsilon\right) \geq 1-\delta,$$

where

$$\epsilon = \sqrt{\frac{G^2\log(1/\delta)}{2n}}.$$

Combining the bounds from the Wasserstein distance and Hoeffding's inequality, we obtain:

$$\mathbb{E}_{(\mathbf{z},\mathbf{m})\sim\mu_t\times\mu_m}[\ell(h,\mathbf{z},\mathbf{m})] \leq \mathbb{E}_{(\mathbf{z},\mathbf{m})\sim\mu_z\times\mu_m}[\ell(h,\mathbf{z},\mathbf{m})] + KW_1(\mu_t,\mu_z)$$

$$\leq \frac{1}{n}\sum_{i=1}^{n}\ell(h,z_i,m_i) + \sqrt{\frac{G^2\log(1/\delta)}{2n}} + KW_1(\mu_t,\mu_z).$$

$$\square$$

## B  ATTACK TYPES

In order to enhance the robustness of the watermark, we consider seven representative types of attack, each applying different types of image distortions. Figure 4 shows the examples of all attack types, and Table 3 presents the attack intensity ranges.

## C  TEST PROMPTS

The evaluation of SAT-LDM's generalization capabilities is conducted using a diverse set of image prompts generated across 10 distinct categories. These categories encompass a wide array of styles, enabling us to rigorously assess the effectiveness of the watermarking method across different image types. The kind of prompt generated is described as follows:

Table 3: **The intensity ranges of attack during training.**

| Attack type | Gaussian blur | Gaussian noise | Brightness | Contrast | Desaturation | Perspective warp | JPEG |
|---|---|---|---|---|---|---|---|
| Intensity range | Kernel size: $7\times 7$ | std: [0, 0.08] | [0, 0.3] | [0.5, 1.5] | [0, 1] | [0, 0.1] | QF: [0, 50] |

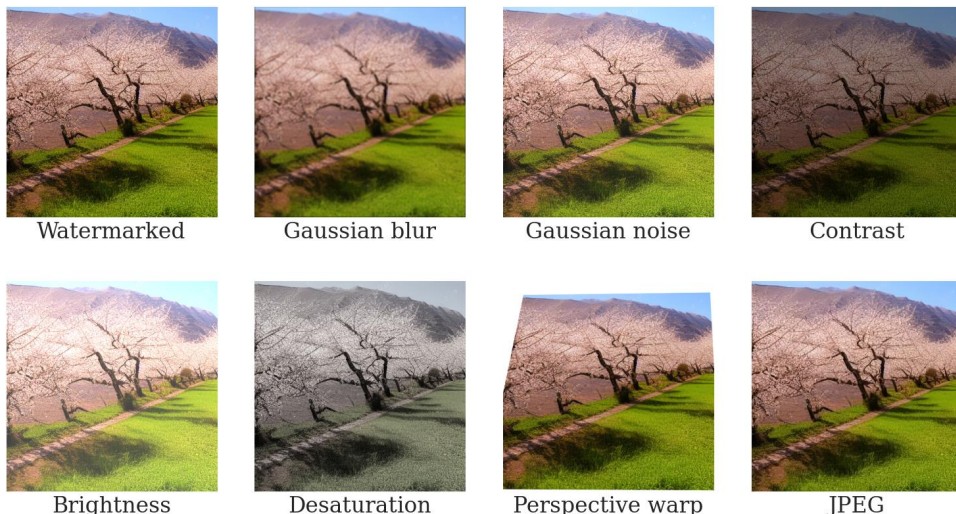

Figure 4: **Illustration of attack types.**

1. Natural landscapes: including mountains, beaches, forests, deserts, and other natural sceneries.

2. Urban landscapes: skylines of different cities, street views, nightscapes, etc.

3. Traditional art styles: style imitations of artistic movements such as the renaissance, baroque, and impressionism.

4. Modern and abstract art: abstract expressionism, cubism, futurism, etc.

5. Everyday objects: common household items, tools, decorations, etc.

6. Technology and futurism: focusing on actual technological advancements and potential future developments such as smart cities, robots, and high-tech equipment.

7. Animals and wildlife: animals in different environments, such as in the wild, zoos, or domestic settings.

8. Historical and cultural events: major historical events, cultural festivals, traditional costumes, etc.

9. Fantasy and science fiction: emphasizing fantasy elements such as magic, mythical creatures, alien worlds, and space exploration.

10. Food and beverages: creative presentations of various dishes, desserts, and drinks.

## D  PSEUDOCODE FOR SAT-LDM

The pseudocode in this section outlines the core steps involved in the training process for the SAT-LDM watermarking framework. The algorithm integrates the watermark embedding process within the latent diffusion model (LDM) and applies a self-augmented training mechanism to enhance the model's generalization capabilities. Specifically, the watermarking module $G_m$ and the message extractor $T_m$ are trained together. During each training epoch, the process involves sampling noise, generating the latent variable through the denoising process, and embedding the watermark into the image. Additionally, adversarial attacks are applied with increasing intensity to simulate real-world conditions and improve robustness. The training loss is minimized by balancing the image reconstruction loss and the message extraction accuracy, ensuring that the watermarked images maintain high quality while the embedded message remains recoverable under various conditions.

Algorithm 1 provides a step-by-step representation of this process, illustrating how the model is iteratively improved through self-augmented training.

---

**Algorithm 1** SAT-LDM with Self-Augmented Training

---

**Input:** Pretrained LDM (Denoising process U, VAE encoder E and VAE decoder D); Watermarking Module $D_m$; Message Extractor $T_m$; Number of Message Bits $L$; Learning rate $\alpha$; Loss weights $\lambda_m$ and $\lambda_I$; Number of epochs $N$; Attack decay coefficient $\gamma_\phi$.
**Output:** Trained watermarking module $G_m$ and message extractor $T_m$.

1: **for** each training epoch $t = 1, ..., N$ **do**
2:     Sample a noise vector $\epsilon \sim \mathcal{N}(0, I)$
3:     Sample a message $\mathbf{m} \sim \{0, 1\}^L$
4:     Generate latent variable $\mathbf{z} = U(\text{""}, \epsilon)$
5:     **Watermark Embedding:**
6:     Generate watermarked image $\mathbf{I}_w = D_m(\mathbf{z}, \mathbf{m})$
7:     **Image Reconstruction Loss:**
8:     Compute original image $\mathbf{I}_o = D(\mathbf{z})$
9:     Compute image loss $\ell_I\left(\mathbf{I}_o, \mathbf{I}_w, \lambda_I\right)$ by Eq. (12)
10:     **Watermark Extraction:**
11:     Apply random attack $\mathbf{I}'_w = \phi(\mathbf{I}_w)$
12:     Extract message $\mathbf{m}' = T_m(\mathbf{I}'_w)$
13:     Compute message loss $\ell_m\left(\mathbf{m}, \mathbf{m}', \lambda_m\right)$ by Eq. (13)
14:     **Total Loss:**
15:     Compute total loss $\mathcal{L} = \ell_I + \ell_m$ in Eq. (9)
16:     **Backpropagation:**
17:     Update parameters of $T_m$ and message processor in $G_m$ using AdamW optimizer
18:     Gradually increase attack intensity by updating $\gamma_\phi$
19: **end for**

---

# E    ADDITIONAL EXPERIMENTAL ANALYSIS

## E.1    DETECTION AND IDENTIFICATION RESULTS

In this section, we consider two common tasks in image watermarking: watermark detection and user identification, to investigate the impact of different training distributions on watermarking performance in practical scenarios.

**Watermark Detection.** The goal of the watermark detection task is to determine whether specific watermark information exists in an image. Given an image $\mathbf{I}$, the detection process involves decoding and verifying the extracted watermark $\mathbf{m}'$ to assess whether it matches the expected watermark. The key challenge in this task lies in efficiently distinguishing between watermarked and non-watermarked images while maintaining accuracy under various image attack scenarios.

**User Identification.** The user identification task not only requires detecting the presence of a watermark in an image but also accurately identifying the embedded watermark message to trace it back to the corresponding user. This task typically involves locating the specific message embedded in an image from a pool of users. As the number of users increases, the complexity of identification rises significantly, especially when the user pool becomes larger.

We refer to the evaluation protocols used in Tree-Ring(Wen et al., 2024), WAVE(An et al.), and WaDiff(Min et al., 2024). Specifically, for the watermark detection task, we use 1,000 watermarked images and 1,000 non-watermarked images to compute the area under the curve (AUC) of the receiver operating characteristic (ROC) curve and the True Positive Rate at a False Positive Rate of 0.001%, denoted as T@0.001%F. For the user identification task, we evaluate our method using user pools of varying sizes, ranging from 10,000 to 1,000,000 users. For each user pool, we randomly select 1,000 users and generate 5 images per user, resulting in a total of 5,000 images. Identification accuracy is then calculated based on these watermarked images.

As shown in Table 4 and Table 5, both "External" and "Free" demonstrate strong performance in the watermark detection and user identification tasks, with no significant differences observed between the two approaches.

Table 4: **Impact of training distributions on watermark detection.**

| Training distributions | AUC/T@0.001%F | | | | | | | | |
|---|---|---|---|---|---|---|---|---|---|
| | None | 1 | 2 | 3 | 4 | 5 | 6 | 7 | Adv. |
| External | 1.000/1.000 | 1.000/0.997 | 1.000/1.000 | 1.000/1.000 | 1.000/1.000 | 1.000/1.000 | 0.995/0.960 | 1.000/1.000 | 0.999/0.995 |
| Free | 1.000/1.000 | 0.995/0.958 | 1.000/1.000 | 1.000/1.000 | 1.000/1.000 | 1.000/0.997 | 1.000/0.999 | 1.000/0.999 | 0.999/0.994 |

Table 5: **Impact of training distributions on user identification.**

| Training distributions | Trace $10^4$/Trace $10^6$ | | | | | | | | |
|---|---|---|---|---|---|---|---|---|---|
| | None | 1 | 2 | 3 | 4 | 5 | 6 | 7 | Adv. |
| External | 1.000/1.000 | 0.994/0.990 | 1.000/1.000 | 1.000/1.000 | 1.000/1.000 | 1.000/1.000 | 0.956/0.930 | 1.000/0.999 | 0.994/0.990 |
| Free | 1.000/0.999 | 0.965/0.957 | 0.999/0.999 | 0.999/0.999 | 0.999/0.999 | 0.997/0.994 | 0.992/0.988 | 0.997/0.992 | 0.994/0.991 |

## E.2 RESULTS ON DIFFERENT PRETRAINED MODELS

We compared the effects of different training distributions on SDv1.5 & v2.1. Table 6 shows the results. We observe that models trained with free distributions significantly improve the quality of watermarked images while maintaining high robustness on both SDv1.5 and SDv2.1.

## E.3 EXPERIMENTAL ANALYSIS WITH LAION-400M AS THE TEST DISTRIBUTION

To ensure the performance improvements are more general, we replace the test data in the "Training Distributions" of Section 5.3 with prompts from LAION-400M (which differs from the training data) and repeat the remaining steps, obtaining Table 7 and Figure 5.

Comparing Tables 1b and 7, we find that after replacing the test data with LAION-400M, the differences in PSNR, FID, and $W_1$ between "External" and "Free" decrease. This is reasonable because, in this experimental setup, the external data comes from LAION-400M. Replacing the test data with prompts from LAION-400M reduces the disparity between the training and test distributions, thereby decreasing $W_1$, which results in higher image quality (PSNR and FID).

On the other hand, in Figures 3 and 5, although there appears to be significant overlap between the "External" and "Test" regions, this is expected, as the generative capacity of the model is derived from external data. The key takeaway from Figure 3 and 5 is the noticeable divergence between the "External" and "Test" distributions in their non-overlapping regions. These divergent regions in "External" likely correspond to samples outside the model's generative capacity, thus introducing noise and limiting generalization when used for training.

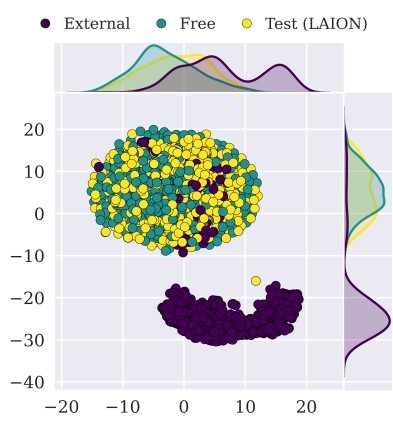

Figure 5: **The t-SNE visualization of proxies for external data, free, and conditional/test distributions (Prompts from LAION-400M).**

While our testing scenarios may not be entirely bias-free, these additional experiments and results further reinforce the robustness and practical relevance of the proposed method across diverse distributions.

## E.4 DOES THE WATERMARKING MODULE ADAPT TO THE NEGATIVE OUTPUT OF UNCONDITIONAL GENERATION?

When generating images, SD employs the noise prediction formula:

$$\epsilon_{\text{guided}} = \epsilon_{\text{uncond}} + w \cdot (\epsilon_{\text{cond}} - \epsilon_{\text{uncond}}),$$

Table 6: **Impact of training distributions on different pretrained models.**

| Pretrained models | Training distributions | PSNR↑ | SSIM↑ | FID↓ | Bit accuracy↑ None | Adv. | AUC/T@0.001%F↑ None | Adv. | Trace $10^4$/Trace $10^5$/Trace $10^6$ ↑ None | Adv. |
|---|---|---|---|---|---|---|---|---|---|---|
| SD v1.5 | External | 37.46 | 0.987 | 3.50 | 1.000 | 0.982 | 1.000/1.000 | 0.999/0.995 | 1.000/1.000/1.000 | 0.994/0.992/0.990 |
| | Free | 40.58 | 0.983 | 2.40 | 1.000 | 0.988 | 1.000/1.000 | 0.999/0.994 | 1.000/1.000/0.999 | 0.994/0.993/0.991 |
| SD v2.1 | External | 36.07 | 0.988 | 4.22 | 1.000 | 0.980 | 1.000/1.000 | 0.995/0.989 | 1.000/1.000/1.000 | 0.985/0.983/0.981 |
| | Free | 41.76 | 0.995 | 2.65 | 1.000 | 0.971 | 1.000/1.000 | 1.000/0.994 | 1.000/1.000/1.000 | 0.990/0.987/0.982 |

Table 7: **Comparison between training distributions (External v.s. Free) and the Wasserstein distance ($W_1$) between training and testing distributions (Prompts from LAION-400M).**

| Training distributions | PSNR↑ | SSIM↑ | FID↓ | Bit accuracy↑ None | 1 | 2 | 3 | 4 | 5 | 6 | 7 | Adv. | $W_1$ ↓ |
|---|---|---|---|---|---|---|---|---|---|---|---|---|---|
| External | 38.87 | 0.988 | 2.89 | 1.000 | 0.973 | 0.999 | 1.000 | 1.000 | 1.000 | 0.935 | 0.967 | 0.982 | 898.6 |
| Free | 41.30 | 0.982 | 2.21 | 0.998 | 0.967 | 0.989 | 0.993 | 0.993 | 0.989 | 0.970 | 0.956 | 0.980 | 669.4 |

where $w$ is the guidance scale.

Since in all cases where the guidance scale is greater than 1, the final output probability is inversely proportional to the unconditional distribution, it raises the possibility that the watermarking module adapts to the negative output of unconditional generation, rather than the proposed method accurately reflecting the generated distribution of diffusion models.

Table 8 presents experimental results with a guidance scale of 1, i.e., direct conditional generation. For the "Free" approach, when the guidance scale is reduced from 7.5 to 1—shifting the test distribution from a mixture of conditional and unconditional distributions to a purely conditional distribution—the FID increases from 2.4 to 3.75 but remains relatively low. In contrast, for the "External" approach, the FID rises from 3.5 to 6.75. These observations indicate the following:

• The "Free" approach does not simply depend on the negative output of the unconditional generation but can adapt to different generation conditions.

• For the "Free" approach, lowering the guidance scale to 1 essentially reduces the diffusion model's reliance on the free/unconditional distribution, resulting in generated images that may slightly deviate from the training distribution. While we assume that the distribution of latent representations generated by all prompts via conditional sampling (conditional generation distribution) is equivalent to that without a specific prompt, this assumption holds only under the ideal condition of sufficiently diverse prompts. Therefore, the slight increase in FID is acceptable and aligns with our proposed Theorem 1.

### E.5 PERFORMANCE OF TRAINING USING LAION-400M CAPTIONS AS PROMPTS

Table 9 compares the results of three different training distributions: 1) 30K images from LAION-400M (LAION Image), 2) 30K images generated using the captions from 1) as prompts (LAION Prompt), and 3) 30K images from the free generation distribution (Free).

Compared to "LAION Image", "LAION Prompt" shows slight improvements in PSNR and FID but still falls short of "Free". This could be due to the inherent bias in prompts, similarly to that in images. Increasing the number of images or prompts might help mitigate this bias, but such an approach would require substantial computational resources and time, and could raise concerns regarding data privacy and copyright.

Although using the unconditionally generated distribution may seem less meaningful, it offers a simpler and more general way to approximate the model's generative capabilities, even when the dataset used by the generative model is unknown.

Table 8: **The impact of training distribution when guidance scale is 1 and 7.5.**

| Guidance scales | Training distributions | PSNR↑ | SSIM↑ | FID↓ | Bit accuracy↑ | |
|---|---|---|---|---|---|---|
| | | | | | None | Adv. |
| 1 | External | 36.71 | 0.987 | 6.75 | 1.000 | 0.987 |
| | Free | 40.90 | 0.984 | 3.75 | 1.000 | 0.994 |
| 7.5 | External | 37.46 | 0.987 | 3.50 | 1.000 | 0.982 |
| | Free | 40.58 | 0.983 | 2.40 | 1.000 | 0.988 |

Table 9: **Performance of training using LAION-400M captions as prompts.**

| Training distributions | PSNR↑ | SSIM↑ | FID↓ | Bit accuracy↑ | | | | | | | | |
|---|---|---|---|---|---|---|---|---|---|---|---|---|
| | | | | None | 1 | 2 | 3 | 4 | 5 | 6 | 7 | Adv. |
| LAION Image | 37.46 | 0.987 | 3.50 | 1.000 | 0.974 | 0.999 | 1.000 | 1.000 | 1.000 | 0.929 | 0.975 | 0.982 |
| LAION Prompt | 38.82 | 0.992 | 3.32 | 1.000 | 0.986 | 0.998 | 1.000 | 1.000 | 1.000 | 0.938 | 0.984 | 0.986 |
| Free | 40.58 | 0.983 | 2.40 | 1.000 | 0.981 | 0.995 | 0.998 | 0.998 | 0.994 | 0.980 | 0.968 | 0.988 |

### E.6 THE IMPACT OF GUIDANCE SCALE ON DISTRIBUTION VISUALIZATION RESULTS

In fact, lots of research has demonstrated that using classifier guidance significantly affects image quality and its alignment with the training prompt, depending on the guidance scale (Ho & Salimans, 2022). (Moreover, generation under guidance is fundamentally different from unconditional generation, the latter often resulting in blurry or content-less outputs.) Therefore, it is counter-intuitive to assume a direct alignment between free and conditional distributions. In the context of this study, it is particularly interesting to investigate how the guidance scale influences our distribution visualization results. Following the setup described in 5.3, we vary only the guidance scale ($gs$) value and visualize the corresponding distributions.

As shown in Figure 6, when $gs \leq 10$, there is almost no noticeable difference between the "Free" and "Test" distributions. However, at $gs = 14$ and $gs = 18$, some differences become apparent. This phenomenon may result from excessively high guidance scales ($gs = 14$ and $gs = 18$), which amplify the guidance signal, causing a certain degree of distributional deviation. On the other hand, as seen in the guidance scale results of Table 2, when $gs$ changes from 10 to 18, there are slight indications that both the FID score and average watermark robustness may degrade. Watermarking methods typically aim to achieve Pareto optimality between watermark image quality and robustness. In our case, the observed distributional deviation might shift this Pareto frontier (either one of the objectives worsens, or both do simultaneously). Nevertheless, even under extreme conditions, such as $gs = 18$, the results remain exceptionally good.

**Why training with free distribution works?** Our method might capture some deeper shared features between conditional and free/unconditional distributions. For instance, both are derived from denoising Gaussian noise. While the denoising processes are not identical, they might retain certain shared features. These features, while meaningless to humans, could be meaningful to the model. As a result, watermarking modules trained on unconditional distributions can generalize effectively to conditional distributions or their hybrids.

## F ADDITIONAL VISUAL COMPARISONS

We provide additional watermarked examples for the prompts discussed in Section C, shown in Figure 7.

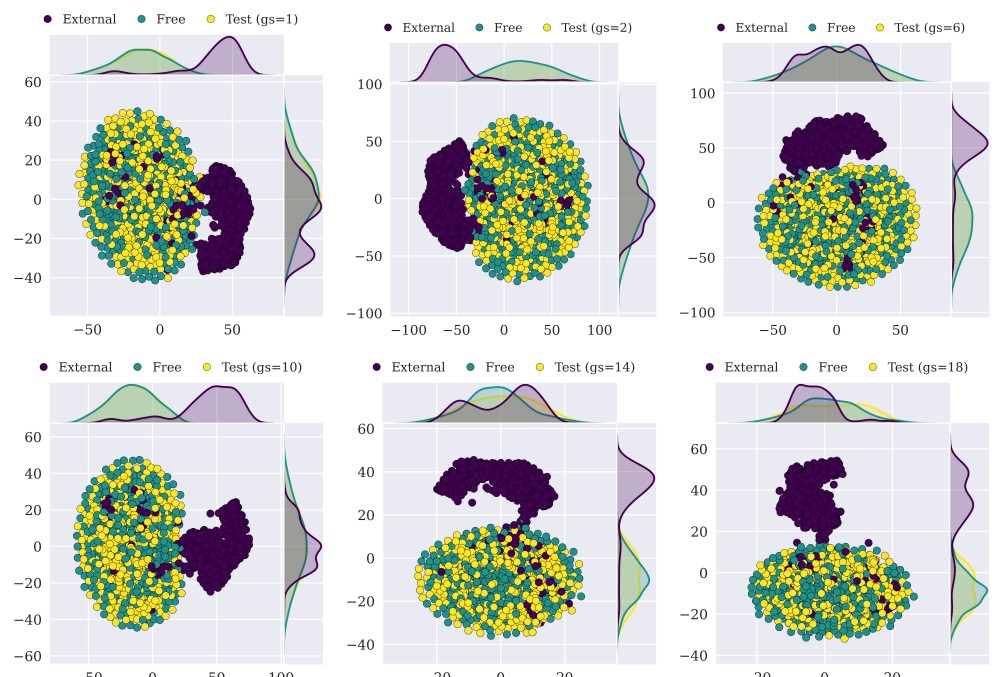

Figure 6: **Distribution visualization results with different guidance scale ($gs$).** Due to the stochastic nature of the t-SNE, the visualization of the distributions may be slightly different.

## G    DISCUSSION ON LIPSCHITZ CONTINUITY OF LOSS FUNCTION

**Detailed Explanation of Lipschitz Continuity Assumption.**    Lipschitz continuity ensures that the loss function does not change too abruptly with respect to its inputs, providing a bounded relationship between changes in predictions and changes in loss. Many widely used loss functions satisfy this property when their inputs are confined to bounded domains (e.g., image data or binary vectors in our case). For instance:

- Mean Squared Error (MSE) Loss: Defined as $L(y, \hat{y}) = (\hat{y} - y)^2$, the MSE loss is Lipschitz continuous when predictions $\hat{y}$ are restricted to a closed interval $[a, b]$. Within this range, the gradient $\nabla L = 2(\hat{y} - y)$ remains bounded, ensuring Lipschitz continuity.
- Cross-Entropy Loss: This loss is Lipschitz continuous when prediction probabilities $\hat{y}$ are confined to the interval $(0, 1)$, guaranteeing bounded gradients.

Additionally, activation functions commonly used in neural networks, such as ReLU and Sigmoid, are inherently Lipschitz continuous. Regularization techniques applied to model weights further help in preventing the Lipschitz constant $K$ from becoming excessively large. Therefore, the assumption is not overly restrictive and can be satisfied by carefully selecting the loss functions, model architectures, and training strategies.

**Estimation of the Lipschitz Constant $K$.**    Estimating the Lipschitz constant of neural networks is an active area of research. For instance, Fazlyab et al. proposed a convex optimization-based approach to efficiently estimate the upper bound of a neural network's Lipschitz constant (Fazlyab et al., 2019). Additionally, Latorre et al. employed polynomial optimization techniques to compute tight upper bounds for $K$ (Latorre et al., 2020). These methods provide both theoretical support and practical tools for estimating $K$.

Empirical methods complement these theoretical approaches by analyzing gradient norms and employing spectral norm analysis on validation datasets. Regularization techniques, such as weight decay and spectral normalization Miyato et al. (2018), not only aid in controlling $K$ but also enhance the generalization capabilities of the model.

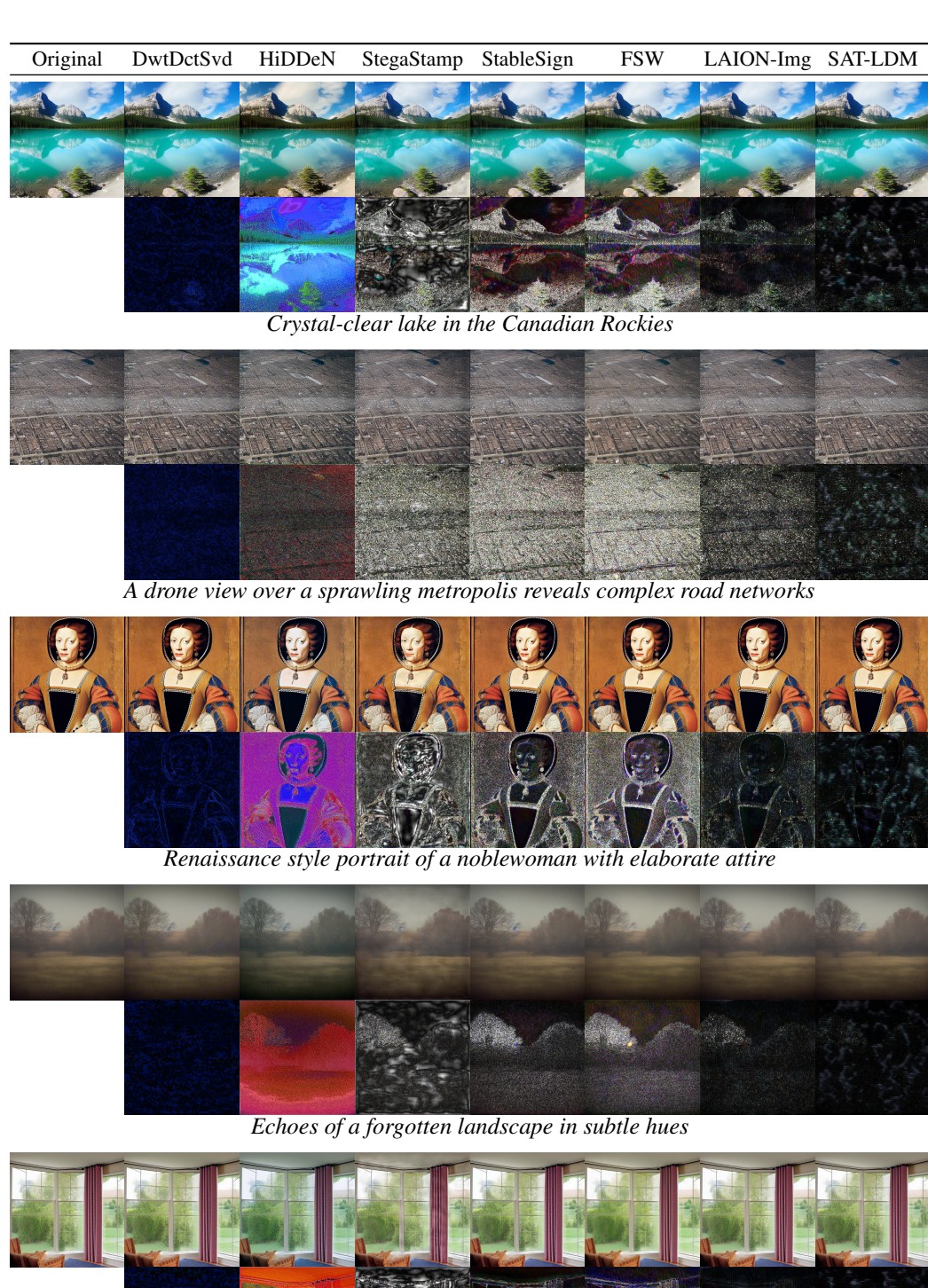

Figure 7: **Watermarked images generated with given prompts.** In each image group, the second row of images is the pixel-wise difference (×10) between the watermarked and non-watermarked images. LAION-Img represents the "External" approach in Section 5.3. Zoom in for better view.

| Original | DwtDctSvd | HiDDeN | StegaStamp | StableSign | FSW | LAION-Img | SAT-LDM |
|----------|-----------|--------|------------|------------|-----|-----------|---------|

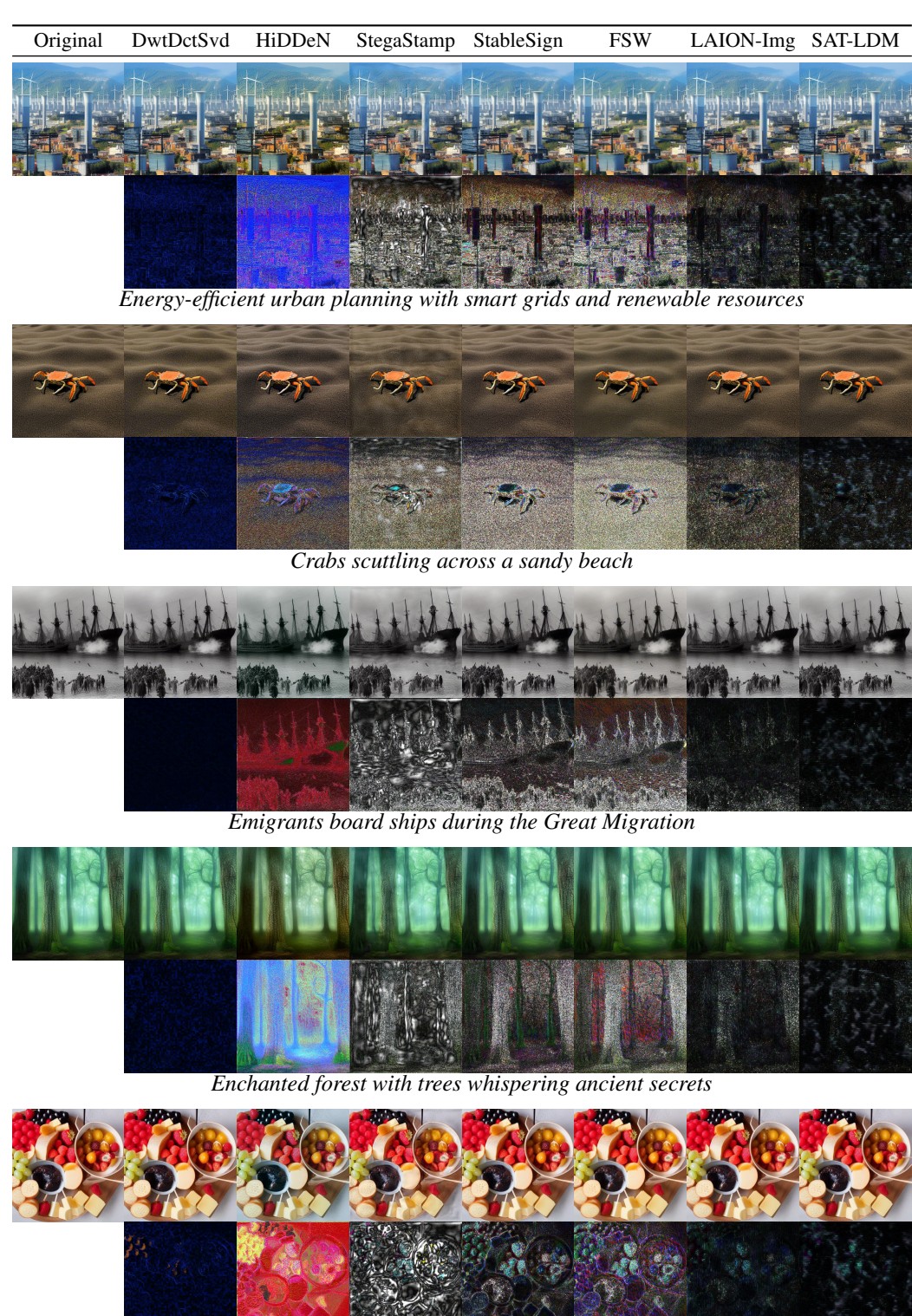

*Energy-efficient urban planning with smart grids and renewable resources*

*Crabs scuttling across a sandy beach*

*Emigrants board ships during the Great Migration*

*Enchanted forest with trees whispering ancient secrets*

*A pot of fondue with assorted fruits and cakes for dipping*

Figure 7: (continued)

