# OpenReview forum: "SAT-LDM: Provably Generalizable Image Watermarking for Latent Diffusion Models with Self-Augmented Training"
_ICLR.cc/2025/Conference — Submitted to ICLR 2025_

### Official Review · Reviewer_gLwe · 2024-10-22

**Soundness:** 2
**Presentation:** 2
**Contribution:** 3
**Rating:** 5
**Confidence:** 2

**Summary:**

The paper observed a mismatch in the target image distribution to be watermarked in diffusion models with previous methods, i.e., external dataset (training phase) vs. generated dataset (testing phase). To address this, the authors propose improving the process by training the watermark using data generated by the diffusion model itself. This enhancement leads to improved performance in the authors' experimental settings.

**Strengths:**

1. The observation that there is a mismatch between the training and testing phases is crucial and can greatly inform future research.
2. The theoretical section is informative.

**Weaknesses:**

Below are the main concerns:

1. The biggest concern stems from the experimental setting. The authors claim that under 1,000 AI-generated prompts, the proposed method surpasses the baseline. However, this number of prompts is far too small for general case measurements, especially considering potential bias in the language models used. Additionally, this comparison seems unfair. The authors should provide a comparison of results using selected prompts from LAION-400M (but not used during training). The reported advantage may solely come from the prompt distribution shift between GPT-generated prompts and those from LAION-400M. Since the proposed method forgoes all prompt information during training (“using empty prompt”), it avoids this shift and may appear better (due to bias from GPT-generated prompts).
2. Another crucial drawback lies in the explanation of why the method works. The authors use generated data with an empty prompt and claim that this somehow represents the conditional distribution of diffusion models, showcasing a demo experiment in Sec. 5.3 Training Distributions. However, this demo experiment needs further verification. Fig. 3 can be interpreted as $d(z_{\text{laion-prompt}}, z_{\text{gpt-prompt}}) > d(z_{\text{no-prompt}}, z_{\text{gpt-prompt}})$, which does not provide any evidence that $d(z_{\text{real-prompt}}, z_{\text{no-prompt}})$ is small in most cases. In fact, it is widely believed that there is a large distance between the conditional and unconditional distributions in DMs, which is why we use (either classifier or classifier-free) guidance.
3. Another critical point is the "guidance" aspect. The authors need to show experimental results with guidance scale = 1, i.e., direct conditional generation (without incorporating an unconditional DM). I am concerned that the real mechanism behind the method's success is that the watermarking modules adapt to the negative output of the unconditional generation. Since in all cases with guidance scale > 1, the final output probability is inversely proportional to the unconditional distribution, this might be the actual reason for the results, rather than the claim that the method accurately reflects the generated distribution of diffusion models.
4. The paper claims two improvements: training data is shifted from a dataset to generated data, and the model structure is updated in Sec. 4.2. However, it is unclear which of these contributes to the observed improvements without an ablation study. There is a risk that the gains come from the structural update rather than the proposed training data change.

Here are some additional minor drawbacks, though they do not significantly impact my overall assessment of the paper:

1. While the intuition—using generated distribution instead of training data—is clear, the writing makes this idea more complicated than necessary. While formal theory is important, the explanation could be simplified to better emphasize the intuition, with theory introduced later.
2. In LaTeX, it would be better to use ``" for double quotes instead of "".

In summary, the method might work (under the specific experiment setting) only due to the fact that the prompt distance between the zero-prompt and GPT-generated ones is closer than the distance between LAION-400M prompts and GPT-generated ones. Given that the results are measured under GPT-generated prompts, it remains unclear how promising the method truly is. Additionally, the explanation for why the proposed method works based on the DM mechanism is insufficient.

Admittedly, the idea of using the generated distribution of DMs for watermarking is potentially promising. However, using the unconditional generated distribution seems less meaningful. It might be better to use conditional generated distributions with diverse prompts, such as those from LAION-5B.

**Questions:**

The questions are connected to the points above, respectively:

1. Can the authors provide results using LAION-400M prompts during testing to ensure the performance improvement is more general?
2. Can the authors offer further insights regarding Fig. 3, such as whether there exists a good subset of LAION-400M prompts that generate images similar to those from GPT-generated prompts? From Fig. 3, there appears to be significant overlap between the External and Test regions. If so, this suggests the proposed testing scenarios may be biased, given that the external training set prompts are more diverse.
3. Could the authors provide additional results and discussion on guidance?
4. Could there be any ablation study, as previously mentioned, to differentiate between the contributions of the structural and training data improvements?

---

> ### Author Response · Authors · 2024-11-26
> **To Reviewer gLwe**
>
> Thank you for your time and dedication to our paper! We have addressed your concerns below and revised the paper to incorporate the reviewers' suggestions. Please let us know if you have further questions.
>
> ---
>
> > **Q1:** The authors claim that under 1,000 AI-generated prompts, the proposed method surpasses the baseline. However, this number of prompts is far too small for general case measurements, especially considering potential bias in the language models used.
>
> **A1:** Due to the high computational cost, we chose 1,000 prompts for comparison, consistent with prior works such as *FSW*[1], *Tree-Ring*[2], and *Gaussian Shading*[3]. Additionally, we have conducted experiments on two new tasks—watermark detection and user identification (Section E.1)—with generated image sizes of 1,000 and 5,000, respectively. These experiments further demonstrate the effectiveness of our method.
>
> ---
>
> > **Q2:** Additionally, this comparison seems unfair. The authors should provide a comparison of results using selected prompts from LAION-400M (but not used during training). The reported advantage may solely come from the prompt distribution shift between GPT-generated prompts and those from LAION-400M. Since the proposed method forgoes all prompt information during training (“using empty prompt”), it avoids this shift and may appear better (due to bias from GPT-generated prompts).
>
> **A2:** Thank you for pointing this out. We would like to emphasize that Table 1 already includes comparisons across diverse datasets: COCO, LAION-400M, Diffusion Prompts, and AI-Generated Prompts. Importantly, the prompts used during testing are independent of those in training, ensuring both robustness and fairness in the evaluation.
>
> ---
>
> > **Q3:** Fig. 3 can be interpreted as $ d(z_{\text{laion-prompt}}, z_{\text{gpt-prompt}}) > d(z_{\text{no-prompt}}, z_{\text{gpt-prompt}}) $, which does not provide any evidence that $ d(z_{\text{real-prompt}}, z_{\text{no-prompt}}) $ is small in most cases.
>
> **A3:** To clarify, the interpretation should be $ d(z_{\text{laion-image}}, z_{\text{gpt-prompt}}) > d(z_{\text{no-prompt}}, z_{\text{gpt-prompt}}) $, as the training distributions consist of images from LAION and free distributions.
> Regarding the specific concern that $ d(z_{\text{real-prompt}}, z_{\text{no-prompt}})$ is small in most cases, we provide additional analysis in Section E.3 (see A13) where we replace GPT-generated prompts with real-world LAION prompts, yielding similar trends and strengthening the robustness of our claims. This approach, while not exhaustive, offers a practical approximation to the diversity of real-world settings and further corroborates the effectiveness of our proposed method.
>
> ---
>
> > **Q4:** In fact, it is widely believed that there is a large distance between the conditional and unconditional distributions in DMs, which is why we use (either classifier or classifier-free) guidance.
>
> **A4:** We would like to clarify that our theoretical assumption focuses on the similarity between the composite distribution of all prompts and the free/unconditional distribution, rather than on the specific conditional distribution to the unconditional distribution.

---

> > ### Author Response · Authors · 2024-11-26
> > **To Reviewer gLwe — Part II**
> >
> > > **Q5:** Another critical point is the "guidance" aspect. The authors need to show experimental results with guidance scale = 1, i.e., direct conditional generation (without incorporating an unconditional DM). I am concerned that the real mechanism behind the method's success is that the watermarking modules adapt to the negative output of the unconditional generation. Since in all cases with guidance scale > 1, the final output probability is inversely proportional to the unconditional distribution, this might be the actual reason for the results, rather than the claim that the method accurately reflects the generated distribution of diffusion models.
> >
> > **A5:** Table 8 presents experimental results with a guidance scale of 1, corresponding to direct conditional generation. For the "Free" approach, when the guidance scale is increased from 7.5 to 1—shifting the test distribution from a mixture of conditional and unconditional distributions to a purely conditional distribution—the FID increases from 2.4 to 3.75 but remains relatively low. In contrast, for "External" approach, the FID rises from 3.5 to 6.75. This indicates the following:
> > (1) The "Free" approach does not solely rely on the negative output of the unconditional generation but can adapt to different generation conditions.
> > (2) For the "Free" approach, lowering the guidance scale to 1 effectively reduces the diffusion model's dependence on the free/unconditional distribution, resulting in generated images that may slightly deviate from the training distribution. While we assume that the distribution of latent representations generated by all prompts via conditional sampling (the conditional generation distribution) is equivalent to that without a specific prompt, this assumption holds only under the ideal condition of sufficiently diverse prompts. Therefore, the slight increase in FID is acceptable and aligns with our proposed Theorem 1.
> >
> > | Guidance scales | Training distributions | PSNR ↑ | SSIM ↑ | FID ↓ | Bit Accuracy ↑ (None) | Bit Accuracy ↑ (Adv.) |
> > |------------------|-------------------------|--------|--------|-------|-----------------------|-----------------------|
> > | 1                | External                | 36.71  | 0.987  | 6.75  | 1.000                 | 0.987                 |
> > |                  | Free                    | 40.90  | 0.984  | 3.75  | 1.000                 | 0.994                 |
> > | 7.5              | External                | 37.46  | 0.987  | 3.50  | 1.000                 | 0.982                 |
> > |                  | Free                    | 40.58  | 0.983  | 2.40  | 1.000                 | 0.988                 |
> >
> > ---
> >
> > > **Q6:** The paper claims two improvements: training data is shifted from a dataset to generated data, and the model structure is updated in Sec. 4.2. However, it is unclear which of these contributes to the observed improvements without an ablation study. There is a risk that the gains come from the structural update rather than the proposed training data change.
> >
> > **A6:** In fact, the "External" results in Table 1(b) can be viewed as the outcome of structural modifications alone, while the "Free" reflects the additional effect of replacing the distribution. Since the authors of FSW provided only model parameters without the training code, we replicated their results based on their paper and made modifications accordingly. Furthermore, it is important to emphasize that we do not consider structural modifications as our main contribution. Our contribution is more theoretical, and thus we focus primarily on the impact of the training distribution.
> >
> > ---
> >
> > > **Q7:** While the intuition—using generated distribution instead of training data—is clear, the writing makes this idea more complicated than necessary. While formal theory is important, the explanation could be simplified to better emphasize the intuition, with theory introduced later.
> >
> > **A7:** We appreciate this feedback and have revised Section 4 for clarity in the updated manuscript.
> >
> > ---
> >
> > > **Q8:** In LaTeX, it would be better to use ``" for double quotes instead of "".
> >
> > **A8:** Thank you for the suggestion; we have made the necessary revisions.
> >
> > ---
> >
> > > **Q9:** Given that the results are measured under GPT-generated prompts, it remains unclear how promising the method truly is.
> >
> > **A9:** See A2 for explanation.
> >
> > ---
> >
> >
> > > **Q10:** Additionally, the explanation for why the proposed method works based on the DM mechanism is insufficient.
> >
> > **A10:** See A5 for explanation.

---

> > > ### Author Response · Authors · 2024-11-26
> > > **To Reviewer gLwe — Part III**
> > >
> > > > **Q11:** Using the unconditional generated distribution seems less meaningful. It might be better to use conditional generated distributions with diverse prompts, such as those from LAION-5B.
> > >
> > > **A11:** Table 9 compares the results of three different training distributions:
> > > 1) 30K images from LAION-400M (LAION Image),
> > > 2) 30K images generated using the captions from 1) as prompts (LAION Prompt),
> > > 3) 30K images from the free generation distribution (Free).
> > >
> > > | Training distributions | PSNR ↑ | SSIM ↑ | FID ↓ | None ↑ | 1 ↑    | 2 ↑    | 3 ↑    | 4 ↑    | 5 ↑    | 6 ↑    | 7 ↑    | Adv. ↑ |
> > > |-------------------------|--------|--------|-------|-------|-------|-------|-------|-------|-------|-------|-------|-------|
> > > | LAION Image            | 37.46  | 0.987  | 3.50  | 1.000          | 0.974 | 0.999 | 1.000 | 1.000 | 1.000 | 0.929 | 0.975 | 0.982 |
> > > | LAION Prompt           | 38.82  | 0.992  | 3.32  | 1.000          | 0.986 | 0.998 | 1.000 | 1.000 | 1.000 | 0.938 | 0.984 | 0.986 |
> > > | Free                   | 40.58  | 0.983  | 2.40  | 1.000          | 0.981 | 0.995 | 0.998 | 0.998 | 0.994 | 0.980 | 0.968 | 0.988 |
> > >
> > > Compared to "LAION Image", "LAION Prompt" shows slight improvements in PSNR and FID but still falls short of "Free". This could be due to the inherent bias in prompts, similarly to that in images. Increasing the number of images or prompts might help mitigate this bias, but such an approach would require substantial computational resources and time, and could raise concerns regarding data privacy and copyright. Although using the unconditionally generated distribution may seem less meaningful, it offers a simpler and more general way to approximate the model's generative capabilities, even when the dataset used by the generative model is unknown.
> > >
> > > ---
> > >
> > > > **Q12:** Can the authors provide results using LAION-400M prompts during testing to ensure the performance improvement is more general?
> > >
> > > **A12:** See A2 for explanation.
> > >
> > > ---
> > >
> > > > **Q13:** Can the authors offer further insights regarding Fig. 3, such as whether there exists a good subset of LAION-400M prompts that generate images similar to those from GPT-generated prompts? From Fig. 3, there appears to be significant overlap between the External and Test regions. If so, this suggests the proposed testing scenarios may be biased, given that the external training set prompts are more diverse.
> > >
> > > **A13:** Overlap is expected, as the model’s generative capability is derived from external data. The key takeaway from Fig. 3 is the noticeable divergence between the "External" and "Test" distributions in their non-overlapping regions. These divergent regions in the "External" distribution correspond to samples beyond the model's generative capacity, thereby introducing noise and limiting generalization when used for training.
> > >
> > > Indeed, as with any experimental design, testing scenarios may inherently contain biases. To address this, we constructe GPT-4-generated prompts spanning diverse styles and complexities for testing. Additionally, we replace the test data in the “Training Distributions” of Section 5.3 with prompts from LAION-400M (which differs from the training data) and repeate the remaining steps. The results, presented in Table 7 and Figure 5, lead to similar conclusions. While our testing scenarios may not be entirely bias-free, these additional experiments and results further reinforce the robustness and practical relevance of the proposed method across diverse distributions.
> > >
> > > | Training distributions | PSNR ↑ | SSIM ↑ | FID ↓ | None ↑ | 1 ↑   | 2 ↑   | 3 ↑   | 4 ↑   | 5 ↑   | 6 ↑   | 7 ↑   | Adv. ↑ | $W_1$ ↓ |
> > > |-------------------------|--------|--------|-------|------|------|------|------|------|------|------|------|------|---------|
> > > | External                | 38.87  | 0.988  | 2.89  | 1.000| 0.973| 0.999| 1.000| 1.000| 1.000| 0.935| 0.967| 0.982| 898.6   |
> > > | Free                    | 41.30  | 0.982  | 2.21  | 0.998| 0.967| 0.989| 0.993| 0.993| 0.989| 0.970| 0.956| 0.980| 669.4   |
> > >
> > > ---
> > >
> > > > **Q14**: Could the authors provide additional results and discussion on guidance?
> > >
> > > **A14**: See A5 for explanation.
> > >
> > > ---
> > >
> > > > **Q15**: Could there be any ablation study, as previously mentioned, to differentiate between the contributions of the structural and training data improvements?
> > >
> > > **A15**: See A6 for explanation.
> > >
> > > ---
> > >
> > > **References:**
> > > [1] Xiong, C., et al. "Flexible and Secure Watermarking for Latent Diffusion Model." ACM MM 2023.
> > > [2] Wen, Y., et al.  "Tree-rings watermarks: Invisible fingerprints for diffusion images." NeurIPS 2023.
> > > [3] Yang, Z., et al. "Gaussian Shading: Provable Performance-Lossless Image Watermarking for Diffusion Models." CVPR 2024.

---

> > ### Comment · Reviewer_gLwe · 2024-11-27
> > **Further review about the paper**
> >
> > Thank you for the new results and I believe some of these indeed make the claim in the paper much more solid. I believe most of the concerns are from writing rather than the experiment or method.
> >
> > 1. I think it is do proper to include main result in the motivation part in Fig 1. But it would maybe clearer to put aggregated result in Fig 1 and detailed result in experiment part. I do miss these details without the notice of the rebuttal as I'm looking for the result comparison only in Experiment Section.
> >
> > 2. About Q3 and Q4, with the additional experiments, I become more confused about the experiment setting. Is the distribution measured by generated images with guidance scale 1 or a larger value such as 7.5 (which is the most common setting for SD generation)? It is only briefly mentioned in L463:
> >
> > > we sampled 1K instances from both the external and free generation distributions.
> >
> > Would guidance scale lead to different distribution visualization results? Indeed, with strong visualization results, it could be accepted that using unconditional generation to replace conditional one. Still, this seems to not be fully achievable.
> >
> > In fact, lots of research has shown that using classifier guidance, the image quality is even largely influenced, as well as its match with the training prompt considering different guidance scales [1]. (And generation is totally different from unconditional one, where the latter usually shows as blurring or content-less generation.) So it is counter-intuitive that one can directly match free and conditional distributions.
> >
> > [1] Ho J, Salimans T. Classifier-free diffusion guidance[J]. arXiv preprint arXiv:2207.12598, 2022.
> >
> > The result with guidance scale=1 and ablation study are promising. These concerns are well addressed.
> >
> >
> > About the training prompt distribution, as the prompt used for prompting GPT-4 is mannually designed, there (should) exists some ablation study about the prompt also. Though I believe this may partially out of the scope of the paper and would accept current result without such ablation study. But the limitation needs to be stated (to inform further study).
> >
> >
> > Overall, I think the results are pretty good (after some clarification in rebuttal). I hope the author could revise the paper by better claiming the results they want to highlight.  I have increased my score to 5 and would appreciate any further discussion about the data distribution discussion, which I believe is truly interesting.

---

> > > ### Author Response · Authors · 2024-11-29
> > > **To Reviewer gLwe**
> > >
> > > We sincerely thank you for your valuable suggestions and for raising the score! We have addressed your new concerns below and have revised the paper accordingly based on your feedback.
> > >
> > > ---
> > >
> > > > **Q1:** It would maybe clearer to put aggregated result in Fig 1 and detailed result in experiment part.
> > >
> > > **A1:** We appreciate the reviewer’s suggestion. Moving Table 1 to the experimental section would indeed enhance the clarity and accessibility of the results. However, implementing this change would require restructuring the figures and related content, which entails a considerable amount of work. Due to the time constraints imposed by the PDF modification policy, we plan to incorporate this adjustment in the camera-ready version.
> > >
> > > ---
> > >
> > > > **Q2:** About Q3 and Q4, with the additional experiments, I become more confused about the experiment setting. Is the distribution measured by generated images with guidance scale 1 or a larger value such as 7.5 (which is the most common setting for SD generation)?
> > >
> > > **A2:** To align with practical use cases, we use a guidance scale of 7.5, unless stated otherwise. We appreciate you pointing out this oversight in our paper. In response, we have clarified this in Section 5.3 by adding the statement: *"Unless stated otherwise, the experimental setup follows the description provided in Section 5.1."*
> > >
> > > ---
> > >
> > > > **Q3:** Would guidance scale lead to different distribution visualization results? Indeed, with strong visualization results, it could be accepted that using unconditional generation to replace conditional one. Still, this seems to not be fully achievable.
> > > In fact, lots of research has shown that using classifier guidance, the image quality is even largely influenced, as well as its match with the training prompt considering different guidance scales. (And generation is totally different from unconditional one, where the latter usually shows as blurring or content-less generation.) So it is counter-intuitive that one can directly match free and conditional distributions.
> > >
> > > **A3:** Following the setup described in Section 5.3, we vary only the guidance scale ($gs$) value and visualize the corresponding distributions, as shown in Figure 6 in Appendix E.6.
> > > As illustrated in Figure 6, when $gs \leq 10$, there is almost no noticeable difference between the “Free” and “Test” distributions. However, at $gs=14$ and $gs=18$, distinct differences emerge. This could be attributed to the excessively high guidance scales ($gs=14$ and $gs=18$), which amplify the guidance signal and lead to certain degrees of distributional deviation. Moreover, as seen in Table 2, when $gs$ changes from 10 to 18, there is a slight degradation in both the FID score and average watermark robustness. Watermarking methods typically aim to achieve Pareto optimality between watermark image quality and robustness. In our case, the observed distributional deviation may shift this Pareto frontier, affecting one or both objectives. Nevertheless, even under extreme conditions like $gs$ = 18, the results remain exceptionally robust.
> > > Our method may capture deeper shared features between conditional and free/unconditional distributions. For instance, both are derived from denoising Gaussian noise. While the denoising processes differ, certain shared features might be preserved. These features, while not interpretable by humans, could still hold meaningful information for the model. As a result, watermarking modules trained on unconditional distributions generalize effectively to conditional distributions.
> > >
> > > ---
> > >
> > > > **Q4:** About the training prompt distribution, as the prompt used for prompting GPT-4 is manually designed, there (should) exist some ablation study about the prompt also. Though I believe this may partially out of the scope of the paper and would accept current result without such ablation study. But the limitation needs to be stated (to inform further study).
> > >
> > > **A4:** This is indeed an important point that warrants further clarification. Thank you for your suggestion! We have added relevant remarks regarding this limitation in the conclusion section, highlighting it as a direction for future research.
> > >
> > > ---
> > >
> > > Once again, we deeply appreciate your patience and highly constructive suggestions! If you have any further questions or concerns, we would be grateful if you could let us know. Moreover, if you find our response satisfactory, we kindly ask you to consider the possibility of improving your rating. Thank you very much for your valuable contribution.

---

> > > > ### Comment · Reviewer_gLwe · 2024-12-02
> > > > **Score remains but confidence decreases**
> > > >
> > > > The reviewer's update mostly convince me, but the claim of
> > > >
> > > > > the similarity between the composite distribution of all prompts and the free/unconditional distribution
> > > >
> > > > is still too strong for me. It may require diverse proof like even the feature for the latent space of the clip in SD to give solid argument. But I am not sure this should be within the scope of the paper as the paper has already provide some proof for this, and current paper's proof may already be enough for a paper about watermark in diffusion models instead of purely exploring properties of dms. So I decide to reduce my confidence to 2 but remain my score to 5.

---

> > > > > ### Author Response · Authors · 2024-12-02
> > > > > **Further clarification to reviewer gLwe**
> > > > >
> > > > > Thank you very much for your patience and timely responses throughout the review process. We greatly appreciate your detailed feedback. However, we believe there might still be some misunderstandings, and we would like to clarify them further.
> > > > >
> > > > > ---
> > > > >
> > > > > > **Q1:** It may require diverse proof like even the feature for the latent space of the clip in SD to give solid argument.
> > > > >
> > > > > **A1:** Our hypothesis focuses on the denoised latent embeddings. Thus, directly analyzing *"the feature for the latent space of the clip in SD"* might not be as relevant or intuitive, as the complete denoising process involves multiple "clip"-assisted denoising steps. In contrast, analyzing the distribution of the denoised latent embeddings directly is more straightforward, which is what we have conducted. Specifically, we have performed extensive t-SNE visualizations on these denoised embeddings and also supported this assumption indirectly with a Wasserstein metric analysis. These experiments collectively enforce the feasibility of our assumptions in practical scenarios.
> > > > >
> > > > > ---
> > > > >
> > > > > > **Q2:** The reviewer's update mostly convince me, but the claim of *“the similarity between the composite distribution of all prompts and the free/unconditional distribution”* is still too strong for me.
> > > > >
> > > > > **A2:** Regarding our hypothesis, ideally, $p\left(\mathbf{z} \mid \mathbf{\epsilon}\right) = \sum p\left(\mathbf{z} \mid \mathbf{\epsilon}, \mathbf{x}^\text{prompt}\right) p(\mathbf{x}^\text{prompt})$, which essentially computes the conditional probability distribution by marginalizing over the prompt variable $\mathbf{x}^\text{prompt}$. This assumption is intuitive and aligns with similar idea presented in prior work, such as [1].
> > > > >
> > > > > ---
> > > > >
> > > > > Thank you once again for acknowledging the contributions of our work! We kindly ask you to reconsider this aspect one last time. If our response have addressed your concerns and you find it reasonable, we would be sincerely grateful if you could reevaluate your score.
> > > > >
> > > > > Regardless of your decision, we deeply appreciate your valuable feedback and guidance, which have greatly enhanced the quality of our work. Reviewers like you make this submission process an enriching and rewarding experience for us.
> > > > >
> > > > > ---
> > > > >
> > > > > **References:**
> > > > > [1] Lu, Y., et al. "Prompt Distribution Learning." CVPR 2022.

---

### Official Review · Reviewer_U5Eu · 2024-10-23

**Soundness:** 3
**Presentation:** 3
**Contribution:** 3
**Rating:** 6
**Confidence:** 2

**Summary:**

The paper proposes an image watermarking scheme that generalizes across different image styles. The authors prove the generalization bound for the watermarked image generator and the message extractor and compare their method against several state-of-the-art approaches in terms of image quality and robustness to removal attacks.

Please use this review sparingly.

**Strengths:**

Authors provide a theoretical guarantee on the generalization error of their approach that, to my knowledge, has not been done in the previous works. Experimental evaluation demonstrates that the proposed method yields watermarked images of better quality than of the competitor's works. The robustness to removal attacks is on the level of state-of-the-art methods.

**Weaknesses:**

The assumption on the Lipschitz continuity of the loss function is somewhat strong: 1) how can one check if it is true and 2) estimate the Lipschitz constant K (what can be quite large leading to unsatisfactory large upper bound)

**Questions:**

I am willing to increase my score if authors provide a detailed explanation on the assumption of Lipschitz continuity of the loss function. How to verify an assumption? Does it hold in practice?

---

> ### Author Response · Authors · 2024-11-26
> **To Reviewer U5Eu — Part I**
>
> Thank you for your time and thoughtful dedication to our paper! We have addressed your concerns below and revised the paper to incorporate the reviewers' suggestions. Please let us know if you have further questions.
>
> ---
>
> > **Q1:** The assumption on the Lipschitz continuity of the loss function is somewhat strong:
> 1) How can one check if it is true?
> 2) How can one estimate the Lipschitz constant $K$, which may be large and lead to an unsatisfactory upper bound?
>
> **A1:**
> 1) **Verifying Lipschitz Continuity:**
> Many common loss functions, including Mean Squared Error (MSE) and Cross-Entropy Loss, are Lipschitz continuous when predictions $\hat{y}$ are constrained to bounded domains. For example, the MSE loss $L(y, \hat{y}) = (\hat{y} - y)^2$ is Lipschitz continuous when $\hat{y}$ is restricted to a closed interval $[a, b]$, as the gradient $\nabla L = 2(\hat{y} - y)$ is bounded within this range. Similarly, the Cross-Entropy Loss is Lipschitz continuous when prediction probabilities $\hat{y}$ are confined to $(0, 1)$, ensuring bounded gradients.
> Additionally, activation functions commonly employed in neural networks, such as ReLU and Sigmoid, are proven to be Lipschitz continuous in prior works [1].
>
> 2) **Estimating the Lipschitz Constant $K$:**
> Estimating the Lipschitz constant of neural networks is an active area of research. For instance, Fazlyab et al. proposed a convex optimization-based approach to efficiently estimate the upper bound of a neural network's Lipschitz constant [2]. Additionally, Latorre et al. employed polynomial optimization techniques to compute tight upper bounds for $K$ [3]. These methods provide both theoretical support and practical tools for estimating $K$.
> ---
>
> > **Q2:** Provide a detailed explanation on the assumption of Lipschitz continuity of the loss function.
>
> **A2:**  For most commonly used loss functions (e.g., MSE loss, Cross-Entropy Loss) [1,4,5], their variations in the parameter space are generally smooth and continuous. Within the bounded data range (e.g., image data or binary vectors in our case), these loss functions are Lipschitz continuous. Furthermore, the fundamental linear mappings and activation functions (e.g., Sigmoid and ReLU) [6,7] used in neural networks are also Lipschitz continuous. Additionally, weights in neural networks are often regularized to prevent overfitting [8,9], which help ensure that the constant $K$ does not grow excessively large. Thus, this assumption is not overly restrictive and can be satisfied by carefully selecting the loss functions, model architectures, and training strategies.
>
> ---
>
> > **Q3:** How to verify an assumption?
>
> **A3:** Verifying the Lipschitz continuity assumption can be approached both theoretically and empirically:
>
> 1) **Theoretical Verification:**
>    - **Function Composition:** Analyze the composition of functions within the model. If each individual function is Lipschitz continuous with known constants, the overall Lipschitz constant can be derived based on the composition rules.
>    - **Bounded Inputs and Parameters:** Ensure that the inputs to the loss function and the model parameters are bounded. Boundedness, combined with Lipschitz continuous activation functions, simplifies the verification of the overall Lipschitz continuity.
>
> 2) **Empirical Verification:**
>    - **Gradient Norms:** Compute the gradient norms of the loss function with respect to inputs over a validation dataset. Consistently bounded gradient norms provide empirical evidence supporting Lipschitz continuity.
>    - **Spectral Norm Analysis:** Utilize techniques like spectral normalization to empirically bound the Lipschitz constant $K$. By normalizing the spectral norms of weight matrices, we can effectively control and estimate $K$.

---

> ### Author Response · Authors · 2024-11-26
> **To Reviewer U5Eu — Part II**
>
> > **Q4:** Does it hold in practice?
>
> **A4:** By incorporating regularization techniques such as spectral normalization [10] and weight decay, we can empirically maintain the Lipschitz constant $ K $ within a manageable range, making this assumption both practical and reasonable in real-world scenarios.
>
> ---
>
> **References:**
> [1] Anil, C., Lucas, J., & Grosse, R. (2019). Sorting out Lipschitz function approximation. *arXiv preprint arXiv:1903.03252*.
> [2] Fazlyab, M., Morari, M., & Pappas, G. J. (2019). Efficient and accurate estimation of Lipschitz constants for deep neural networks. *NeurIPS 2019*.
> [3] Latorre, F., Lodi, A., & Martello, S. (2020). Lipschitz constant estimation for Neural Networks via sparse polynomial optimization. *ICLR 2019*.
> [4] Bousquet, O., & Elisseeff, A. (2002). Stability and generalization. *Journal of Machine Learning Research*, 2, 499–526.
> [5] Zhang, C., et al. (2021). Understanding deep learning (still) requires rethinking generalization. *Communications of the ACM*, 64(3):107–115.
> [6] Bartlett, P. L., Foster, D. J., & Telgarsky, M. J. (2017). Spectrally-normalized margin bounds for neural networks. *NeurIPS 2017*.
> [7] Ledoux, M. (2001). The concentration of measure phenomenon. *Mathematical Surveys and Monographs, Vol. 89*.
> [8] Bengio, Y., et al. (2017). Deep learning. *MIT Press*.
> [9] Srivastava, N., et al. (2014). Dropout: A simple way to prevent neural networks from overfitting. *Journal of Machine Learning Research*, 15(56), 1929–1958.
> [10] Takeru, M., et al. (2018). Spectral normalization for generative adversarial networks. *arXiv preprint arXiv:1802.05957*.

---

> > ### Comment · Reviewer_U5Eu · 2024-11-27
> > **Official comment to authors**
> >
> > I appreciate the time and effort the authors put in into answering my questions. I would like to keep my score.

---

### Official Review · Reviewer_iTqJ · 2024-10-29

**Soundness:** 2
**Presentation:** 2
**Contribution:** 3
**Rating:** 5
**Confidence:** 5

**Summary:**

This paper introduces SAT-LDM, a watermarking method integrated within latent diffusion models (LDMs) to generate watermarked images. The authors argue that SAT-LDM improves generalization across image styles without compromising quality, unlike existing methods which reportedly degrade image content.

**Strengths:**

The paper focuses on a significant topic—embedding watermarks in images generated by diffusion models like Stable Diffusion—to address the growing need for copyright protection of AI-generated content.

**Weaknesses:**

- The paper's **motivation lacks clarity and appears misguided.**
The authors claim that current diffusion-native watermarking methods degrade image quality by comparing watermarked images to unwatermarked counterparts generated from the same prompt. This comparison is irrelevant because users of diffusion-native watermarking methods would not see unwatermarked images beforehand. Such comparisons would only apply to post-watermarking methods, where preserving the original image content is necessary.
Additionally, the claim that watermarking performance is compromised across image styles is questionable, as style diversity is largely dictated by the diffusion model's training data rather than the watermarking process itself.

- The paper’s Figure 2 depicts a VAE decoder as frozen while taking message embedding as input. If the authors follow the FSW structure, as stated in Section 4.2, **the figure is incorrect because FSW fuses message embeddings into the UNet-decoder, not the VAE decoder**. If the figure is correct, freezing the entire diffusion model (including the VAE) while training only the message processor and extractor introduces the risk of extracting false-positive messages from non-watermarked images (i.e., images generated without embedding). Thus, **the paper should include a false positive rate (FPR) analysis to evaluate this aspect**.

- The authors claim “no external data” usage in Figure 1, but they must have used training data for the two message components, just like the existing methods.

- The approach lacks innovation, as it mainly replicates FSW’s structure and uses the spatial transformer network from StegaStamp to improve robustness.

-  The experimental evaluation lacks critical ablations and comparisons. The paper should investigate the impact of using different pre-trained diffusion models and evaluate the method under strong attacks, such as diffusion-based attacks gauge robustness under adversarial conditions. WAVES [1] could be a reference in this case.

[1] An, Bang, et al. "WAVES: Benchmarking the Robustness of Image Watermarks." Forty-first International Conference on Machine Learning.

**Questions:**

- What dataset did the authors use to train the message processor and extractor? How does this align with the claim of “no external data”?
- Given the potential for the message extractor to output messages from non-watermarked images, why is there no report on FPR?
- Does Figure 2 accurately represent the proposed structure? If so, how does a frozen VAE decoder process message embeddings, and what are the implications of only training the message processor and extractor?

---

> ### Author Response · Authors · 2024-11-26
> **To Reviewer iTqJ — Part I**
>
> Thank you for your time and thoughtful engagement with our paper! We have addressed your concerns below and revised the paper to incorporate the reviewers' suggestions. Please let us know if you have further questions.
>
> ---
>
> > **Q1:** The authors claim that current diffusion-native watermarking methods degrade image quality by comparing watermarked images to unwatermarked counterparts generated from the same prompt. This comparison is irrelevant because users of diffusion-native watermarking methods would not see unwatermarked images beforehand. Such comparisons would only apply to post-watermarking methods, where preserving the original image content is necessary.
>
> **A1:** We respectfully disagree with the critique of the comparison methodology. The comparison between watermarked and non-watermarked images is designed as a diagnostic tool to quantitatively assess the impact of watermark embedding on image quality. It is not intended to imply that end-users would directly compare these two types of images. Instead, this approach establishes a fair baseline for evaluating the effects of watermarking methods on image quality. By measuring changes introduced by watermarking against an unaltered baseline, our methodology ensures a rigorous assessment. This practice is consistent with prior works, such as *Stable Signature*[1], *FSW*[2], and *WaDiff*[3], which also evaluate methods against non-watermarked counterparts. While we acknowledge that end-users may not encounter non-watermarked images, such comparisons are essential for benchmarking and advancing methods within the research community.
>
> ---
>
> > **Q2:** Additionally, the claim that watermarking performance is compromised across image styles is questionable, as style diversity is largely dictated by the diffusion model's training data rather than the watermarking process itself.
>
> **A2:** This concern appears to arise from a misunderstanding of the relationship between watermarking and image style diversity. While image style diversity is inherently derived from the diffusion model’s training data, the ability of the watermarking module to generalize across image styles is determined by its training strategy and the distribution of its training data. It is important to note that watermarking methods trained on external datasets may exhibit biases when encountering rare or previously unseen styles.
>
> ---
>
> > **Q3:** The paper’s Figure 2 depicts a VAE decoder as frozen while taking message embedding as input. If the authors follow the FSW structure, as stated in Section 4.2, **the figure is incorrect because FSW fuses message embeddings into the UNet-decoder, not the VAE decoder**.
>
> **A3:** Regrettably, this statement is factually incorrect.
> Section 4.3 of the *FSW* [2] paper explicitly states:
> > “In order to achieve the goal of changing embedding message flexibly without training or fine-tuning again, we fuse the message-matrix $m_w$ in **fine-tuned LDM-decoder $D_w$**.”
>
> Furthermore, Section 3.2 specifies:
> > “Note that, **the LDM-decoder used in this work is the variational autoencoder (VAE)**, which is widely used in stable diffusion models.”
>
> As described in the FSW paper, watermark embedding is performed at the VAE decoder stage, which aligns with our implementation. Figure 2 in our paper accurately represents this architecture, where the VAE decoder parameters are kept frozen, and the message embedding is incorporated through a plug-in message processor.

---

> > ### Author Response · Authors · 2024-11-26
> > **To Reviewer iTqJ — Part II**
> >
> > > **Q4:** If the figure is correct, freezing the entire diffusion model (including the VAE) while training only the message processor and extractor introduces the risk of extracting false-positive messages from non-watermarked images (i.e., images generated without embedding). Thus, the paper should include a false positive rate (FPR) analysis to evaluate this aspect.
> >
> > **A4:** Our experiments primary focus on bit accuracy and image quality metrics. However, as you suggested, we have conducted additional False Positive Rate (FPR) analyses in Section E.1. Following the evaluation protocols of *Tree-Ring*[4], *WAVE*[5], and *WaDiff*[3], we calculate the area under the curve (AUC) of the receiver operating characteristic (ROC) curve and the True Positive Rate at a False Positive Rate of 0.001% (T\@0.001%F) using 1,000 watermarked and 1,000 non-watermarked images.
> >
> > | Training Distributions | None       | 1          | 2          | 3          | 4          | 5          | 6          | 7          | Adv.       |
> > |-------------------------|------------|------------|------------|------------|------------|------------|------------|------------|------------|
> > | External               | 1.000/1.000 | 1.000/0.997 | 1.000/1.000 | 1.000/1.000 | 1.000/1.000 | 1.000/1.000 | 0.995/0.960 | 1.000/1.000 | 0.999/0.995 |
> > | Free                   | 1.000/1.000 | 0.995/0.958 | 1.000/1.000 | 1.000/1.000 | 1.000/1.000 | 1.000/0.997 | 1.000/0.999 | 1.000/0.999 | 0.999/0.994 |
> >
> > Our "External" and "Free" approaches demonstrate exceptional performance, achieving average AUC and T\@0.001%F values exceeding 99% even under adversarial conditions (Adv.).
> >
> > ---
> >
> > > **Q5**: The authors claim “no external data” usage in Figure 1, but they must have used training data for the two message components, just like the existing methods.
> >
> > **A5**: We clarify that "no external data" refers to our self-training approach, which utilizes internally generated free distributions (Section 4). In contrast to methods that rely on external datasets (e.g., LAION-400M and COCO), SAT-LDM generates training samples entirely within the LDM’s operational domain, thereby eliminating reliance on external sources.
> >
> > ---
> >
> > > **Q6:** The approach lacks innovation, as it mainly replicates FSW’s structure and uses the spatial transformer network from StegaStamp to improve robustness.
> >
> > **A6:** It is note that **Our contribution lies in the theory**. While we build on the established methods in *FSW*[2] and *StegaStamp*[6], our primary contributions lie in the theoretical framework (generalization bounds) and practical implementation (self-augmented training), which significantly enhance the quality of watermarked images. Reviewer U5Eu also acknowledged this, stating:
> > > “Authors provide a theoretical guarantee on the generalization error of their approach that, to my knowledge, **has not been done in previous works**.”

---

> > > ### Author Response · Authors · 2024-11-26
> > > **To Reviewer iTqJ — Part III**
> > >
> > > > **Q7:** The experimental evaluation lacks critical ablations and comparisons. The paper should investigate the impact of using different pre-trained diffusion models and evaluate the method under strong attacks, such as diffusion-based attacks.
> > >
> > > **A7:** We have included evaluations on Stable Diffusion v2.1 (see Section E.2). Our results demonstrate that models trained with free distributions significantly improve watermarked image quality while maintaining high robustness on both SDv1.5 and SDv2.1 models.
> > >
> > > | Pretrained Models | Training Distributions | PSNR ↑ | SSIM ↑ | FID ↓ | Bit Accuracy ↑ (None) | Bit Accuracy ↑ (Adv.) | AUC/T\@0.001%F ↑ (None) | AUC/T\@0.001%F ↑ (Adv.) | Trace 10⁴/Trace 10⁵/Trace 10⁶ ↑ (None) | Trace 10⁴/Trace 10⁵/Trace 10⁶ ↑ (Adv.) |
> > > |-------------------|-------------------------|--------|--------|-------|-----------------------|-----------------------|-------------------------|-------------------------|-----------------------------------------|-----------------------------------------|
> > > | SD v1.5          | External               | 37.46  | 0.987  | 3.50  | 1.000                 | 0.982                 | 1.000/1.000            | 0.999/0.995            | 1.000/1.000/1.000                     | 0.994/0.992/0.990                     |
> > > |                   | Free                   | 40.58  | 0.983  | 2.40  | 1.000                 | 0.988                 | 1.000/1.000            | 0.999/0.994            | 1.000/1.000/0.999                     | 0.994/0.993/0.991                     |
> > > | SD v2.1          | External               | 36.07  | 0.988  | 4.22  | 1.000                 | 0.980                 | 1.000/1.000            | 0.995/0.989            | 1.000/1.000/1.000                     | 0.985/0.983/0.981                     |
> > > |                   | Free                   | 41.76  | 0.995  | 2.65  | 1.000                 | 0.971                 | 1.000/1.000            | 1.000/0.994            | 1.000/1.000/1.000                     | 0.990/0.987/0.982                     |
> > >
> > > We have conducted comprehensive experiments under diverse settings, including comparisons with other watermarking methods, training distributions, sample sizes, message bit lengths, sampling methods, guidance scales, inference steps, and pretrained models (detailed in Section E.2). Additionally, we also addressed watermark detection and user identification (discussed in Section E.1). These experiments encompassed various attack scenarios (Section 5.1), validating the efficacy of our method and supporting our theoretical claims. Regarding the strong adversarial attacks, due to constraints in time and computational resources, we have deferred their exploration to future work.
> > >
> > > ---
> > >
> > > > **Q8:** What dataset did the authors use to train the message processor and extractor? How does this align with the claim of “no external data”?
> > >
> > > **A8:** See A5 for explanation. The training process uses samples from the LDM’s free generation distribution, without relying on any external datasets.
> > >
> > > ---
> > >
> > > > **Q9:** Given the potential for the message extractor to output messages from non-watermarked images, why is there no report on FPR?
> > >
> > > **A9:** See A4 for explanation.
> > >
> > > ---
> > >
> > > > **Q10:** Does Figure 2 accurately represent the proposed structure? If so, how does a frozen VAE decoder process message embeddings, and what are the implications of only training the message processor and extractor?
> > >
> > > **A10:** In our case, a frozen VAE decoder means its parameters remain fixed, while the intermediate computations are designed to process the message embeddings. The key aspect of training the information processor and the extractor is that we update only these components, leaving all other model parameters unchanged.
> > >
> > > ---
> > >
> > > **References:**
> > >
> > > [1] Fernandez, P., et al. "The stable signature: Rooting watermarks in latent diffusion models.", in CVPR 2023.
> > > [2] Xiong, C., et al. "Flexible and Secure Watermarking for Latent Diffusion Model.", in ACM MM 2023.
> > > [3] Min, R., et al. "A watermark-conditioned diffusion model for ip protection.", in ECCV 2024.
> > > [4] Wen, Y., et al.  "Tree-rings watermarks: Invisible fingerprints for diffusion images.", in NeurIPS 2023.
> > > [5] An, B., et al. "WAVES: Benchmarking the Robustness of Image Watermarks.", in ICML 2024.
> > > [6] Tancik, M., et al. "Stegastamp: Invisible hyperlinks in physical photographs.", in CVPR 2020.

---

> ### Author Response · Authors · 2024-12-02
> **Looking Forward to Further Feedback**
>
> Dear Reviewer iTqJ,
>
> Thank you for taking the time to review our submission and provide your thoughtful feedback. We hope our rebuttal has adequately addressed your concerns. Specifically, we have clarified the misunderstandings, highlighted our main contributions, and provided additional ablation experiments. As the discussion period approaches its end, we kindly request that you review these points and consider updating your evaluation accordingly. Moreover, if you find our response satisfactory, we kindly ask you to consider the possibility of improving your rating.
>
> Thank you very much for your valuable contribution, and we look forward to your response.
>
> Best regards,
> The authors

---

> > ### Comment · Reviewer_iTqJ · 2024-12-02
> > **Thanks for the additional explanation.**
> >
> > Thanks for the explanation and additional experiments. I will increase my score to 5, but still, I don't think this paper is ready to be accepted at this point.
> >
> > I do acknowledge that this paper gives a good theory foundation for their method, self-augmented training, and this aspect should be appreciated. But if we consider this contribution before reading this paper, I would expect this paper will show that 'free' is better than 'external' on different kinds of existing watermarking methods, because actually 'free' (what you proposed) is not really a watermarking method, but a training strategy to improve the generalization ability of the watermarking methods, which is orthogonal to proposing a new watermarking method, and that's why your implementation is highly depended on FSW in this paper. Then the paper structure will be like explaining 'free' is better than 'external' in theory, then proving it by comparing 'free' and  'external' under different existing watermarking methods.
> >
> > Besides, your motivation now comes from 'better generalizing to different image styles when applied to real-world scenarios'. It is kind of tricky to prove such generalization capability. Yes, you proved that the watermarked images have good quality compared to the original image in terms of PSNR and SSIM, those pixel-to-pixel metrics. But there is still a gap between "a good generalization capability" and "a good image quality". I would suggest changing the paper motivation a little bit. For instance, your simple yet effective method can improve the image quality of any existing robust watermarking methods, to achieve the frontier of the tradeoff between the image quality and image watermark robustness.

---

> ### Author Response · Authors · 2024-12-03
> **Further Clarification to Reviewer iTqJ**
>
> Thank you for your detailed feedback. However, we believe there might still be some misunderstandings, and we would like to provide further clarification.
> > **Q1:** actually 'free' (what you proposed) is not really a watermarking method, but a training strategy to improve the generalization ability of the watermarking methods, which is orthogonal to proposing a new watermarking method, and that's why your implementation is highly depended on FSW in this paper.
>
> **A1:** Thank you for your thoughtful analysis of our method. Regarding the positioning of the "free" strategy, our primary goal was to **rethink how watermarking modules in LDM are trained** and propose a training approach that enhances training-based watermarking methods. As you pointed out, this strategy extends beyond watermarking tasks; the same design principle can be applied to other tasks with similar architectures that demand improved generalization capability. We greatly appreciate your insights into the broader potential of our method.
>
> Our decision to adopt the FSW-like structure was driven by the need to control experimental variables, ensuring fairness and clarity in demonstrating the core advantages of the "free" strategy. This choice also aligns with our commitment to scientific rigor and resource constraints. For example, training our method from scratch on FSW-like structure using a single RTX 4090 takes less than half a day on average, whereas Stable Signature requires 8 GPUs for around a full day [1], not including its further fine-tuning for different users. According to the Stable Signature paper, their experiments demanded approximately **2000 days or ≈ 50,000 GPU-hours**. Additionally, Stable Signature requires separate fine-tuning for different users, whereas the more flexible watermarking design like FSW represents a forward-looking trend in this field.
>
> ---
>
> > **Q2:** Besides, your motivation now comes from 'better generalizing to different image styles when applied to real-world scenarios'. It is kind of tricky to prove such generalization capability. Yes, you proved that the watermarked images have good quality compared to the original image in terms of PSNR and SSIM, those pixel-to-pixel metrics. But there is still a gap between "a good generalization capability" and "a good image quality".
>
> **A2:** First, we would like to clarify that we have never claimed, nor attempted to demonstrate that, "good generalization ability" and "good image quality" are equivalent. Rather, our definition of "a good generalization capability" explicitly includes **image watermark robustness**. A robust watermarking method should embody both attributes; therefore, we believe it is essential to address both aspects, rather than overlooking one.
>
> Second, it appears your concern pertains to the gap between theory and practice in our work. However, we argue that this gap does not signify a disconnect, but rather reflects the inherent challenges of aligning local experimental metrics with the complexities of real-world data. In our theoretical analysis, we employed the **Wasserstein distance** to demonstrate that the "free" generation distribution mitigates the discrepancy between training and testing distributions, providing theoretically supports for improved generalization.
>
> In our experiments, we used **watermarked image quality metrics** and **watermark robustness** to assess generalization across different prompts and image styles as **proxy indicators** of **generalization ability**. While these metrics may not fully capture all facets of generalization, they are interpretable, practical and relevant for real-world applications. Furthermore, we designed experiments across diverse data distributions (e.g., COCO, LAION-400M, Diffusion Prompts, and AI-generated prompts) to reflect real-world generalization performance. This combination of theoretical analysis and empirical evaluation strengthens our argument of improved generalization.
>
> ---
>
> Thank you once again for acknowledging the contributions of our work! As the rebuttal process is coming to a close, we would appreciate if you could let us know if you have any further concerns and/or consider raising the score.
>
>
> **References:**
> [1] Fernandez, P., et al. "The stable signature: Rooting watermarks in latent diffusion models.", in CVPR 2023.

---

### Official Review · Reviewer_UfTs · 2024-11-04

**Soundness:** 4
**Presentation:** 4
**Contribution:** 4
**Rating:** 6
**Confidence:** 4

**Summary:**

This paper studies how to watermark diffusion models across diverse image styles. The authors propose a training-based watermark method SAT-LDM. In particular, the authors plug a message processor into the VAE decoder to obtain watermarked images from latents. During the training, SAT-LDM jointly trains the message processor and message extractor. The diffusion model is fixed during the training. No external data is required for the training. Theoretical analysis is provided to demonstrate the generalization ability of the proposed method. Experiments show that SAT-LDM can generalize across different image styles.

**Strengths:**

- The writing is clear and easy to follow.
- The method does not require additional training data.
- A theoretical guarantee is provided for the proposed method
- Experiments show that the method produces effective watermarks while maintaining high image fidelity.

**Weaknesses:**

- More visualization results, especially of different image styles, can help demonstrate the generalization of the proposed SAT-LDM.

The authors addressed my concerns.
I agree with reviewer iTqJ that the novelty of the proposed method might be relatively limited.
Therefore I will maintain my score.

**Questions:**

Please refer to the weaknesses

---

> ### Author Response · Authors · 2024-11-26
> **To Reviewer UfTs**
>
> Thank you for your time and dedication to our paper! We have addressed your concerns below and revised the paper to incorporate the reviewers' suggestions. Please let us know if you have further questions.
>
> ---
> > **Q1:** More visualization results, especially of different image styles, can help demonstrate the generalization of the proposed SAT-LDM.
>
> **A1:** Thank you for recognizing the significance of our work. In response to your feedback, we have added additional visualization examples in Section F of the revised manuscript. These new results encompass various image styles, as discussed in Section C. By comparing SAT-LDM with different watermarking methods, we further highlight its generalization capability across diverse image styles. We hope these enhancements address your request, and we sincerely appreciate your efforts in helping us improve the quality of our paper.

---

> ### Comment · Reviewer_UfTs · 2024-11-27
> **Thanks for response**
>
> The authors addressed my concerns. Yet I agree with reviewer iTqJ that the novelty of the proposed method might be relatively limited. Therefore, I will maintain my score.

---

> ### Author Response · Authors · 2024-12-03
> **Further Clarification to Reviewer UfTs**
>
> Thank you for your feedback. We appreciate your acknowledgment that our rebuttal effectively addressed your concerns. Notably, **reviewer iTqJ has recognized the theoretical novelty of our proposed method following the rebuttal**. In light of this clarification, we kindly request you to reevaluate your score or opinion.

---

### Author Response · Authors · 2024-11-26
**General Response**

We want to express our gratitude for the valuable comments and constructive feedbacks of all reviewers. We are encouraged that the advantages of our paper were generally appreciated by the reviewers. The reviewers thought *“The writing is clear and easy to follow. A theoretical guarantee is provided for the proposed method. Experiments show that the method produces effective watermarks while maintaining high image fidelity.”* (Reviewer UfTs), *“The paper focuses on a significant topic to address the growing need for copyright protection of AI-generated content.”* (Reviewer iTqJ), *“Authors provide a theoretical guarantee on the generalization error of their approach that, to my knowledge, has not been done in the previous works. The robustness to removal attacks is on the level of state-of-the-art methods.”* (Reviewer U5Eu), *“The observation that there is a mismatch between the training and testing phases is crucial and can greatly inform future research. The theoretical section is informative.”* (Reviewer gLwe).

We have carefully gone through each review and summarize several important questions commonly raised and address them in this thread. We have also uploaded a rebuttal version of the paper with the following revisions (all new texts and contents are in blue, note that this is not the final version):

1. **Additional Visual Comparisons**: We have included diverse visualization examples across different image styles in Appendix Section F.
2. **New Experimental Analyses**:
   - Extended experiments on watermark detection and user identification with larger datasets (e.g., 5,000 images).
   - Results on different pretrained models.
   - Experimental analysis using LAION-400M as the test distribution.
   - Additional experiments conducted with a guidance scale of 1.
   - Evaluation of training performance when using LAION-400M captions as prompts.
3. **Discussion on Lipschitz Continuity of Loss Function**: We have included discussion on lipschitz continuity assumption of the loss function in Appendix Section G, to discuss its rationality and verifiability in practice.

---

### Meta-Review · Area_Chair_FMGE · 2024-12-21

**Metareview:**

2x borderline accept, 2x borderline reject. This paper studies how to watermark diffusion models across diverse image styles by introducing a training-based watermark method that avoids external datasets and provides theoretical guarantees on generalization. The reviewers agree on the (1) clear writing and solid theoretical framing, (2) effectiveness of watermark embedding without reliance on external data, (3) promising performance on image fidelity and robustness, and (4) value of the self-augmented training approach. However, they note (1) limited novelty relative to existing diffusion-native watermarking methods, (2) incomplete analysis of false positives and distribution assumptions, (3) potential mismatch between unconditional and conditional generation in real use cases, and (4) insufficient ablations distinguishing the benefits of new training data from structural changes. The authors have followed up with additional experiments, clarifications on distribution gaps, and new results on FPR and guidance scales, yet several reviewers remain partially unconvinced, so the AC leans to not accept this submission.

**Additional Comments On Reviewer Discussion:**

N/A

---

### Decision · Program_Chairs · 2025-01-22

Reject